# A genome-wide CRISPR/Cas9 knockout screen identifies TMEM239 as an important host factor in facilitating African swine fever virus entry into early endosomes

**Dongdong Shen[1,2©], Guigen Zhang[3©], Xiaogang Weng[4©], Renqiang Liu[1], Zhiheng Liu[5], Xiangpeng Sheng[1], Yuting Zhang[4], Yan Liu[4], Yanshuang Mu[4], Yuanmao Zhu[1], Encheng Sun[1], Jiwen Zhang[1], Fang Li[1], Changyou Xia[1], Junwei Ge[2], Zhonghua Liu[4]\*, Zhigao Bu[1]\*, Dongming Zhao[1]\***

**1** State Key Laboratory for Animal Disease Control and Prevention, National African Swine Fever Para-reference Laboratory, Harbin Veterinary Research Institute, Chinese Academy of Agricultural Sciences, Harbin, China, **2** College of Veterinary Medicine, Northeast Agricultural University, Harbin, China, **3** Institute of Human Virology, Key Laboratory of Tropical Disease Control of Ministry of Education, Zhongshan School of Medicine, Sun Yat-sen University, Guangzhou, China, **4** Key Laboratory of Animal Cellular and Genetics Engineering of Heilongjiang Province, Engineering Research Center of Intelligent Breeding and Farming of Pig in Northern Cold Region, College of Life Science, Northeast Agricultural University, Harbin, China, **5** Peking-Tsinghua Center for Life Sciences, School of Life Sciences, Peking University, Beijing, China

© These authors contributed equally to this work.
\* liuzhonghua@neau.edu.cn (ZL); buzhigao@caas.cn (ZB); zhaodongming@caas.cn (DZ)

## Abstract

African swine fever (ASF) is a highly contagious, fatal disease of pigs caused by African swine fever virus (ASFV). The complexity of ASFV and our limited understanding of its interactions with the host have constrained the development of ASFV vaccines and antiviral strategies. To identify host factors required for ASFV replication, we developed a genome-wide CRISPR knockout (GeCKO) screen that contains 186,510 specific single guide RNAs (sgRNAs) targeting 20,580 pig genes and used genotype II ASFV to perform the GeCKO screen in wild boar lung (WSL) cells. We found that knockout of transmembrane protein 239 (TMEM239) significantly reduced ASFV replication. Further studies showed that TMEM239 interacted with the early endosomal marker Rab5A, and that TMEM239 deletion affected the co-localization of viral capsid p72 and Rab5A shortly after viral infection. An e*x vivo* study showed that ASFV replication was significantly reduced in TMEM239[-/-] peripheral blood mononuclear cells from TMEM239 knockout piglets. Our study identifies a novel host factor required for ASFV replication by facilitating ASFV entry into early endosomes and provides insights for the development of ASF-resistant breeding.

## Author summary

ASFV is a major threat to pig industry worldwide and relies on host factors to complete its replication cycle. We developed a genome-wide CRISPR knockout (GeCKO) screen and

**Data Availability Statement:** All the relevant date are in the manuscript and its supporting information files.

**Funding:** DMZ was supported by Heilongjiang Provincial Natural Science Foundation of China (JQ2023C005) and Innovation Program of Chinese Academy of Agricultural Sciences (CAAS-CSLPDCP-202301). RQL was supported by National Key R&D Program of China (2021YFD1800101). ZGB was supported by National Key R&D Program of China (2019YFE0107300). The funders did not play any role in the study design, data collection and analysis, decision to publish, or preparation of the manuscript.

**Competing interests:** The authors have declared that no competing interests exist.

identified TMEM239, a member of the transmembrane (TMEM) protein family, as a potential host factor required for ASFV replication. TMEM239 interacts with the early endosome marker Rab5A and TMEM239 knockout impedes ASFV entry into early endosomes. We further generated TMEM239 knockout pigs and ASFV replication was significantly reduced in TMEM239$^{-/-}$ peripheral blood mononuclear cells (PBMCs). Our findings will contribute to the development of strategies for pig breeding against ASF.

## Introduction

African swine fever (ASF) is a highly contagious and fatal viral disease of pigs caused by the African swine fever virus (ASFV) [1,2]. ASF infects both domestic and wild pigs across a wide range of pig breeds and ages with high morbidity and mortality [3]. ASF initially appeared in Kenya, Africa, in 1921 [4] and subsequently disseminated swiftly across numerous countries and regions globally [5–7]. The ASF global epidemic poses a serious threat to global food security and livestock health [8]. Antiviral therapeutics or vaccines against ASF are lacking, a predicament that is largely attributed to our limited understanding of ASFV etiology and ASFV-host interactions [9,10].

The life cycle of ASFV commences with viral adsorption and subsequent entry into the host cells. Previous studies revealed that ASFV enters host cells through receptor-mediated endocytosis, necessitating specific conditions including temperature, energy, cholesterol, and low pH [11,12]. Recent studies have demonstrated that ASFV employs both endocytosis and macropinocytosis to enter host cells and completes subsequent infection [13,14]. The endocytic mechanism utilized by this virus primarily involves clathrin-mediated endocytosis (CME), characterized by the receptor-dependent internalization of virions through the formation of a clathrin-coat beneath the plasma membrane [15]. Macropinocytosis is the non-selective process of endocytosing extracellular nutrients and liquid-phase macromolecules by cells [16]. ASFV enters host cells via macropinocytosis, which is closely associated with the cell types and the methods used for preparing the viral stocks [17]. ASFV mainly infects primary porcine monocytes and macrophages. Although cell surface markers such as CD163 and CD45 are reported to be involved in ASFV replication [18, 19], downregulating the expression of these host genes did not prevent ASFV from infecting these cells, suggesting that ASFV entry into host cells involves complex processes.

Viruses utilize the cellular machinery of the host to complete their life cycles, and some host factors play crucial roles in viral replication [20]. As a member of the nucleocytoplasmic large DNA viruses (NCLDV), numerous studies have demonstrated that many host factors participate in ASFV replication. CSF2RA interacts with the ASFV envelope glycoprotein CD2v to inhibit cell apoptosis by regulating the JAK2-STAT3 pathway, thereby promoting viral replication [21]. CD1d serves as a host binding partner for ASFV capsid protein p72 and facilitates ASFV entry via CME [22]. The tyrosine kinase receptor AXL and phosphatidylserine (PS) receptor Tim-4 recognize PS on the surface of ASFV virions to mediate viral entry through macropinocytosis [23,24]. Clustered Regularly Interspaced Short Palindromic Repeats (CRISPR)/Cas9 is an effective technique for introducing targeted loss-of-function mutations at specific genomic loci [25]. The genome-wide CRISPR knockout (GeCKO) screen has proven to be efficient and reliable in identifying host factors necessary for viral replication [26,27]. A GeCKO screen in porcine cells revealed that several genes, including RFXANK, RFXAP, SLA-DMA, SLA-DMB, and CIITA, play important roles in the early stages of ASFV infection [28].

In this study, we performed a genome-wide CRISPR/Cas9 screen in wild boar lung (WSL) cells to identify host factors required for ASFV infection. We found that transmembrane protein 239 (TMEM239) is essential for ASFV infection, and knockout of TMEM239 significantly inhibits ASFV replication and disrupts ASFV entry into the early endosomes. We further established TMEM239 knockout pigs and evaluated ASFV replication in TMEM239[-/-] peripheral blood mononuclear cells (PBMCs). Our findings will contribute to the development of strategies for pig breeding against ASF.

## Methods

### Ethics statement

All experiments with ASFVs were conducted in the enhanced biosafety level 3 (P3) facilities in the Harbin Veterinary Research Institute (HVRI) of the Chinese Academy of Agricultural Sciences (CAAS), which is approved by the Ministry of Agriculture and Rural Affairs of China and the China National Accreditation Service for Conformity Assessment. The protocol for the animal studies was approved by the Committee on the Ethics of Animal Experiments of the HVRI, CAAS.

### Cells and viruses

Wild boar lung (WSL) and HEK-293T cells were cultured in Dulbecco's modified Eagle's medium (DMEM) with 10% fetal bovine serum (FBS; Gibco, USA) and 1% penicillin/streptomycin. Porcine alveolar macrophages (PAMs) and peripheral blood mononuclear cells (PBMCs) were obtained from 4-week-old specific-pathogen-free (SPF) piglets as previously described [29] and were cultured in Roswell Park Memorial Institute 1640 medium (RPMI 1640, Gibco, USA) supplemented with 10% FBS and 1% penicillin/streptomycin. Porcine embryonic fibroblast (PEF) cells were isolated from the 35-day-old fetuses of SPF Large White pigs as previously described [30] and were cultured in DMEM with 15% FBS and 1% penicillin/streptomycin. All cells were maintained at 37°C with 5% $CO_2$.

Wild-type virulent genotype II virus HLJ/18 (WT-HLJ18) and wild-type lower virulent genotype I strain SD/DY-I/21 (WT-SD) were isolated from domestic pigs and stored in our laboratory [7,31]. WSL-adapted viruses, including genotype II strains ad-HRB1 and ad-7GD, were developed in our laboratory. Briefly, ad-HRB1 was derived from the naturally attenuated genotype II strain of HLJ/HRB1/20 (GenBank: MW656282) [32] through 30 consecutive passages in WSL cells, following previously described method [33]. Ad-7GD was derived from 25 consecutive passages of a seven-gene deletion live attenuated vaccine candidate (HLJ/18-7GD) in WSL cells [34]. Detailed information on the above four strains used in this study was shown in S1 Table. The virus stocks of WT-SD, ad-HRB1, and ad-7GD were propagated in WSL cells. In brief, three strains were individually inoculated into WSL cells. Upon observing significant CPE, the cell culture supernatants were collected, and viruses were purified by sucrose density gradient centrifugation, following previously described method [1]. For WT-HLJ18, the strain was propagated in porcine alveolar macrophages (PAMs), and the virus was purified using the same method as above. All virus stocks were aliquoted and stored at -80°C.

### Antibodies

The antibodies including mouse anti-Myc (Product No. M4439, Sigma, USA), mouse anti-Flag (Product No. B3111, Sigma, USA), mouse anti-Rab5A (Product No. ab66746, Abcam, US), mouse anti-Rab7 (Product No. ab137029, Abcam, US), mouse anti-β-Actin (Product No. 66009, Proteintech, USA), mouse anti-Tubulin (Product No. 66031, Proteintech, USA), rabbit

anti-Flag (Product No. 80010, Proteintech, USA), Alexa Fluor 488-labeled goat anti-mouse IgG (H + L) (Product No. A0428, Beyotime, China), and Alexa Fluor 546-labeled goat anti-rabbit IgG (H + L) (Product No. A-11035, Thermo Invitrogen, USA) were purchased from the indicated vendors. Mouse monoclonal anti-p30 and p54 were produced in our laboratory as previously described [35,36]. Rabbit polyclonal anti-p72 was prepared and stored in our laboratory. Briefly, p72 trimer protein was expressed and purified in yeast cells following previous methods [37,38], then mixed with immune adjuvant and immunized rabbits three times. Subsequently, blood was collected from the rabbits and the p72 polyclonal antibody was prepared from the collected serum.

## Construction of plasmids

The cDNA sequences of the porcine TMEM239 gene (Transcript: ENSSSCT00000007842.4) and Rab5A gene (Transcript: ENSSSCT00000045035.3) were synthesized by Genscript (Nan-Jing, China). TMEM239 was fused with Flag or Myc and cloned into the pCAGGS vector (pTMEM239-Flag/pTMEM239-Myc). Rab5A was fused with Myc and cloned into the pCAGGS vector (pRab5A-Myc). SgRNAs targeting TMEM239 and Rab14 were cloned into lentiGuide-Puro (#52963, Addgene), respectively. Eight ASFV genes (*B646L, O61R, E248R, B438L, E183L, D117L, E199L,* and *E120R*) localized in the outer membrane, capsid, or inner membrane were amplified from the ad-HRB1 genomic DNA and cloned into the pCAGGS-C-HA plasmid to obtain viral proteins fusing HA tag at C terminus (pB646L-HA, pO61R-HA, pE248R-HA, pB438L-HA, pE183L-HA, pD117L-HA, pE199L-HA and pE120R-HA), respectively. All primers for gene cloning and sequencing are available in S2 Table.

## Construction of the swine GeCKO screen sgRNA plasmid library

To establish the swine genome-wide CRISPR knockout (GeCKO) screen sgRNA library, 6–10 sgRNAs were designed against each coding gene, lncRNA, or miRNA in Sus scrofa genome 11.1 as described previously [39]. All 186,510 specific sgRNAs targeting 20,580 porcine genes and 1000 non-targeting control sgRNAs (S3 Table) were designed and synthesized by using CustomArray 90 K arrays (Synbio-Tech, China). Then, sgRNAs were amplified by using PCR and cloned into lentiGuide-Puro plasmids by using the golden-gate method (NEBridge Golden Gate Assembly Kit). Products of golden-gate were purified by using a MinElute PCR purification Kit (QIAGEN, Germany) and electrotransformed into Endura electrocompetent cells (Biosearch Technologies, USA). To ensure comprehensive coverage, parallel transformations were performed and colony numbers were counted to reach 500-fold the total number of sgRNAs in the library. Subsequently, the sgRNA library plasmids were extracted by using a Plasmid Plus Maxi kit (QIAGEN, Germany). Then, the sgRNA cassettes from the sgRNA plasmid library were amplified using KOD DNA polymerase (TOYOBO, Japan) as previously described [39]. PCR products were purified by using a MinElute PCR purification Kit (QIAGEN, Germany) and then sequenced by using an Illumina HiSeq2000 platform (Shenzhen Zhong Nong Jing Yue Biotech Company Limited, China). Coverage and homogeneity of sgRNAs were analyzed by using the MAGeCK algorithm [40].

## Construction of WSL-Cas9 cell line

The strategy for constructing a WSL cell line overexpressing Cas9 (WSL-Cas9) is to insert the Cas9 expression cassette into the Rosa26 locus of the cellular genome. The targeting donor vector was constructed as previously described [41]. In brief, the Cas9 expression cassette was amplified from PX458-Lck (#159430, Addgene) and cloned into the PUC57-Blast vector. And

the 512bp 5' homology arm and the 1116bp 3' homology arm amplified by genomic PCR were cloned into the upstream and downstream of the Cas9 expression cassette, respectively (Fig 1A). The sgRNA sequence targeting Rosa26 locus was cloned into lentiGuide-Puro (#52963, Addgene) and then co-transfected into WSL cells with the targeting donor vector. Following blasticidin (Product No. A1113902, Thermo Invitrogen, USA) selection, monoclonal cells overexpressing Cas9 were obtained by using the limiting dilution method, and Cas9 expression was confirmed by using Western blot.

## Generation of the WSL GeCKO screen cell library and host factor screening for ASFV infection

The WSL GeCKO screen cell library was generated by using the lentiGuide-Puro two-vector system for sgRNA delivery as previously described [39]. Briefly, $5.5 \times 10^8$ WSL-Cas9 cells were infected with the genome-wide sgRNA library lentiviruses at an MOI of 0.3 to attain less than one sgRNA per cell and at least 1000 cells for each sgRNA. The cells were screened with 1.5 μg/ml puromycin for two weeks to achieve >95% gene disruption. The GeCKO WSL cell population expanded 3–4-fold during puromycin selection. Subsequently, $1.8 \times 10^8$ WSL GeCKO screen cells were taken from cells proliferating during puromycin screening for deep sequencing to determine sgRNA coverage (S4 Table). And an additional $1.8 \times 10^8$ WSL GeCKO screen cells were infected with ad-7GD at an MOI of 1. Seven days later, the surviving cells were collected and reseeded to expand up to $1.8 \times 10^8$ cells and were then infected with ad-7GD at an MOI of 1. This process was repeated one more time, and then the genomic DNA (gDNA) was extracted from the surviving cells by using the QlAamp DNA Mini Kit (QIAGEN, Germany). The inserted sgRNA cassettes were amplified for deep sequencing. The sequencing data were analyzed by using the MAGeCK algorithm [40], which assesses the statistical significance of candidate gene rankings through the robust ranking aggregation (RRA).

## Generation of TMEM239 and Rab14 knockout cells

The sgRNA sequences targeting TMEM239 and Rab14 were cloned into the plasmid lenti-Guide-puro and transfected into HEK293T cells along with helper plasmids to generate lentivirus. Subsequently, the lentivirus was transduced into WSL-Cas9 cells at a MOI of 0.3 to generate polyclonal CRISPR KO cells. Following puromycin selection, TMEM239-KO and Rab14-KO monoclonal cells were obtained by limiting dilution and confirmed by Sanger sequencing. RT-qPCR (Relative quantitative reverse transcription-PCR) was used to evaluate the KO efficiency of the target genes.

Polyclonal TMEM239-KO cell lines were generated using three different sgRNAs (S2 Table). Similarly, these sgRNAs were cloned into plasmid lentiGuide-puro and transfected into HEK293T cells with helper plasmids to generate lentivirus. Subsequently, the lentivirus was transduced into WSL-Cas9 cells at an MOI of 0.3 to generate polyclonal CRISPR KO cells, respectively. After puromycin selection, the polyclonal TMEM239 KO cell line was obtained.

## Multistep viral growth curves

WSL and TMEM239$^{-/-}$ cells were seeded into 24-well plates at the same density and then infected with ad-7GD, ad-HRB1, and WT-SD at an MOI of 1, respectively. After a 1-h incubation, the cells were maintained in DMEM medium with 2% FBS. The cell culture supernatants were collected at daily intervals until cell death due to the virus infection. Viral growth curves were characterized by using quantification of viral genome copies (p72) and viral titers.

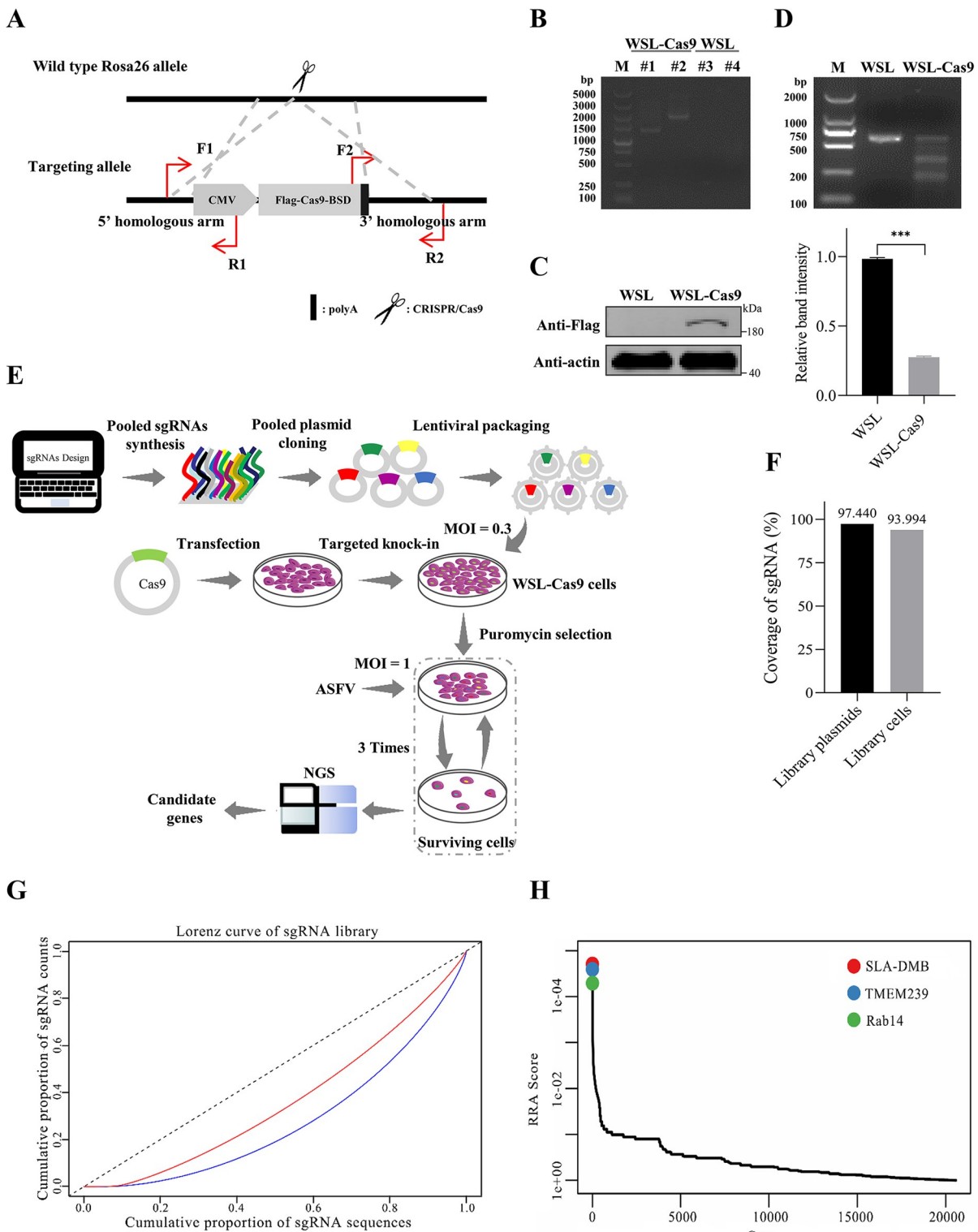

**Fig 1. GeCKO screening for host factors associated with ASFV replication.** (A) Construction of the WSL-Cas9 cell line. The primer pairs F1/R1 and F2/R2 were used to confirm the correct insertion of the Cas9 expression cassette into the Rosa26 locus of the cellular genome. (B) PCR identification of the WSL-Cas9 cell line. Lanes #1 and #2, amplified by the F1/R1 and F2/R2 primers, respectively, were indicative of the presence of WSL-Cas9 cells, with sizes of 1561 bp and 1909 bp, respectively. Lanes #3 and #4 are the amplification results of WSL cells identified by F1/R1 and F2/R2 primer PCR, and the amplification results are negative. M: DL5,000 DNA Marker. (C) Western blot analysis of

Cas9 expression in WSL-Cas9 cells. (D) Validation of nuclease cleavage ability in WSL-Cas9 cells by using the T7 endonuclease 1 (T7E1) cleavage assay. A sgRNA targeting GGTA1 with a well-documented high gene editing efficiency was selected to evaluate the cleavage activity of the Cas9 protein expressed in the WSL-Cas9 cells used in the T7 endonuclease 1 (T7E1) cleavage assay. ***$p < 0.001$. (E) Schematic of the Genome-wide CRISPR/Cas9 knockout screen for host factors associated with ASFV infection. (F) Coverage of CRISPR knockout library plasmids and library cells compared with synthetic sgRNAs lists. (G) Homogeneity of library plasmids and library cells was assessed by using Lorenz curve analysis. (H) Results of the MAGeCK algorithm for candidate gene ranking. Results of the MAGeCK algorithm that assesses the statistical significance of candidate gene rankings through the robust ranking aggregation (RRA).

## siRNA knockdown of the TMEM239 gene

PAMs were transfected with small interfering RNAs (siRNA) targeting TMEM239 or a negative control siRNA (NC-siRNA) at a concentration of 30 nM by using the Lipofectamine RNAiMAX transfection reagent (Invitrogen, USA). All siRNAs were designed and synthesized by GenePharma (Shanghai, China) and listed in S2 Table. The mRNA level of TMEM239 was measured to evaluate the knockdown efficiency by RT-qPCR at 36 h post-transfection (hpt). The siRNA-treated PAMs were then infected with WT-HLJ18 at an MOI of 0.1 at 36 hpt. The cells and supernatants were collected to assess virus replication levels.

## Virus adsorption and internalization assays

WSL and TMEM239$^{-/-}$ cells were infected with ad-HRB1 (MOI = 5) and incubated for 2 h at 4°C. Subsequently, the cells were extensively washed with ice-cold PBS to remove viral particles that had not been adsorbed to the cells, and the cells were harvested for total DNA or protein extraction to quantify adsorbed ASFV virions. To determine the effect of TMEM239 on ASFV internalization, WSL and TMEM239$^{-/-}$ cells were infected with ad-HRB1 (MOI = 5) and incubated for 2 h at 4°C. The cells were extensively washed with ice-cold PBS to remove unbound viral particles and then incubated for 2 h at 37°C to allow the internalization of viral particles followed by a trypsin treatment to remove any remaining surface-bound virus particles. To validate the efficiency of the trypsin treatment in removing surface-bound virus particles, a control group (Con) treated with trypsin was included at the end of the incubation at 4°C. Similarly, cells were harvested for total DNA or protein extraction to quantify internalized ASFV virions.

## Pull-down assay, mass spectrometry, and Co-IP

For the TMEM239 pull-down assays, Flag-tagged TMEM239 plasmid and empty Flag plasmid were transfected into WSL cells, respectively. At 24 hpt, the cells were lysed with ice-cold lysis buffer on a flip shaker for 1 h at 4°C. Cell lysates were centrifuged (12000 $g$/min, 10 min) at 4°C and immunoprecipitated by anti-Flag (M2) agarose beads (Sigma-Aldrich, USA) overnight at 4°C on a roller. The TMEM239-binding proteins in the immunoprecipitants were resolved by using 12% SDS-PAGE and silver staining (Thermo Invitrogen, USA). The gel was cut and processed for liquid chromatography-tandem mass spectrometry (LC-MS/MS).

For TMEM239 and Rab5A co-immunoprecipitation (Co-IP) assay, HEK293T cells were seeded into 6-well plates and transfected with Flag-tagged TMEM239 (pTMEM239-Flag) and Myc-tagged Rab5A (pRab5A-Myc) together or separately. At 36 hpt, the cells were washed with ice-cold PBS and lysed with Pierce IP Lysis Buffer (Thermo Scientific, USA) containing a complete protease inhibitor cocktail (Abcam, USA). Cell lysates were centrifuged (12000 $g$/min, 10 min) at 4°C and incubated with beads conjugated with anti-Myc affinity gel (Sigma-Aldrich, USA) overnight at 4°C on a roller. The incubated beads were washed five times with lysis buffer and the bound proteins were detected by Western blotting. For TMEM239 and eight ASFV proteins co-immunoprecipitation (Co-IP) assay, HEK293T cells were seeded into

6-well plates and transfected with Myc-tagged TMEM239 (pTMEM239-Flag) and HA-tagged viral genes (pB646L-HA, pO61R-HA, pE248R-HA, pB438L-HA, pE183L-HA, pD117L-HA, pE199L-HA and pE120R-HA) together or separately. Subsequently, the viral proteins interacting with TMEM239 were detected following the aforementioned procedure.

## Immunofluorescence microscopy

To detect the formation of viral factories by using immunofluorescence microscopy, WSL and TMEM239$^{-/-}$ cells were seeded into 20-mm dishes and infected with ad-HRB1 at an MOI of 1. At 18 h post-infection (hpi), the cells were washed with PBS, fixed with 4% paraformaldehyde, and permeabilized with 0.25% Triton X-100 for 15 min at room temperature. After being blocked with 0.5% BSA for 1 h, the cells were incubated with a primary antibody against the viral capsid protein p72 for 2 h at room temperature. After being washed with PBST, the cells were incubated with Alexa Fluor 546-labeled goat anti-rabbit IgG (H + L) for 1 h. DAPI staining was used to visualize nuclear and viral DNA, and the fluorescence signals of the samples was analyzed by use of confocal microscopy.

WSL and TMEM239$^{-/-}$ cells were also seeded into 20-mm dishes and incubated with ad-HRB1 (MOI = 20) for 2 h at 4°C. The cells were then washed with ice-cold PBS and incubated with fresh ice-cold medium for 30, 45, and 60 min at 37°C. The cells were washed with ice-cold PBS, fixed with 4% paraformaldehyde, and then permeabilized with 0.25% Triton X-100 for 15 min at room temperature. After being blocked with 0.5% BSA for 1 h, the cells were incubated with primary antibodies against the viral capsid protein p72 and the early endosomal marker Rab5A for 2 h at room temperature. After being washed with PBST, the cells were incubated with Alexa Fluor 488-labeled goat anti-mouse IgG (H + L) and Alexa Fluor 546-labeled goat anti-rabbit IgG (H + L) for 1 h. DAPI staining was used to visualize the nuclei, and the fluorescent signals of the samples was analyzed by use of confocal microscopy.

## Flow cytometry analysis

The frozen PBMCs from wild-type piglets and TMEM239$^{-/-}$ piglets were thawed, washed with RPMI 1640 medium containing 10% FBS, and incubated at 37°C with 5% $CO_2$ for 1 h. Subsequently, the PBMCs were washed with PBS, adjusted to $10^6$ cells/100 μL and resuspended in staining buffer (Thermo Fisher Scientific, USA). The cell populations in PBMCs were analyzed using the following antibodies: Pacific Blue mouse anti-pig CD45 (Product No. MCA1222PB, Bio-Rad, USA), mouse anti-pig CD3ε-FITC (PPT3) (Product No. 4510–02, SouthernBiotech, USA), PE mouse anti-pig CD172a (Monocyte/Granulocyte, Product No. 561499, BD Biosciences, USA), PE mouse anti-pig CD21 (B cells, Product No. MA5-28769, Thermo Fisher Scientific, USA), Alexa Fluor 647 mouse anti-pig CD4 (Product No. 561472, BD Biosciences, USA), and FITC mouse anti-pig CD8 (Product No. 551303, BD Biosciences, USA), according to a previously published method [42].

## Hemadsorption (HAD) assay

The hemadsorption (HAD) assay was conducted following established procedures with minor adjustments [43]. Primary porcine peripheral blood mononuclear cells (PBMCs) were seeded in 96-well plates. Subsequently, the samples and pig red blood cells were then added into the plates and titrated in triplicate using 10 × dilutions. The quantity of ASFV was determined by identification of characteristic rosette formation representing haemadsorption of erythrocytes around infected cells. HAD was observed for 7 days, and 50% HAD doses (HAD$_{50}$) were calculated by using the method of Reed and Muench [44].

## Results

### Establishment of a genome-wide CRISPR/Cas9 screen in WSL cells

To identify host factors required for ASFV infection, the GeCKO screen was performed in the WSL cell line stably expressing Cas9 (WSL-Cas9), which was generated by integrating the Cas9 expression cassette into the Rosa26 locus of the cellular genome (Fig 1A) [45]. The accurate insertion of the Cas9 expression cassette into the Rosa26 locus was confirmed by using PCR (Fig 1B), and Cas9 expression was confirmed by Western blotting (Fig 1C). An sgRNA targeting GGTA1 with a well-documented high gene editing efficiency was selected to evaluate the cleavage activity of the Cas9 protein in WSL-Cas9 cells by using the T7 endonuclease 1 (T7E1) cleavage assay [46], and analysis of the band intensity revealed that Cas9 had more than 70% cleavage activity in WSL-Cas9 cells (Fig 1D). Viral replication kinetics of ASFV, including WSL-adapted strains (ad-7GD and ad-HRB1) and wild-type strain (WT-SD and WT-HLJ18), used in current research were assessed (S1A and S1B Fig). The results indicated that all viruses can reach higher titers when replicating in PAMs. Ad-7GD replicated similarly in WSL cells and WSL-Cas9 cells transduced with lentivirus carrying the gRNA targeting GGTA1, indicating that the insertion of the Cas9 expression cassette and lentivirus transduction had no noteworthy impact on viral replication in WSL cells (S1C and S1D Fig). A schematic of the GeCKO screen used to identify host factors associated with ASFV infection is shown in Fig 1E. The GeCKO screen sgRNAs were designed, synthesized, amplified, and cloned into lentiGuide-Puro plasmids to generate the GeCKO screen sgRNA plasmid library. Deep sequencing revealed that the sgRNA plasmid library had 97.44% coverage of the sgRNA content (Fig 1F) and Lorenz curve analysis indicated that our library was highly homogeneous (Fig 1G). The sgRNA plasmid library was transfected with helper plasmids into HEK293T cells to produce the lentiviral library. Subsequently, the lentiviral library was transduced into WSL-Cas9 cells at a multiplicity of infection (MOI) of 0.3 (Fig 1E), thereby establishing the cell library (WSL-GeCKO cell library). Deep sequencing of sgRNA sequences within the cell library revealed 93.994% coverage of the sgRNA content (Fig 1F), and Lorenz curve analysis indicated a more pronounced degree of bias in the homogeneity of sgRNA distribution compared to the sgRNA plasmid library (Fig 1G). The WSL-GeCKO cell library was infected with ad-7GD at an MOI of 1. The surviving cells were collected and reseeded for two more rounds of virus infection (Fig 1E). Eventually, genomic DNA from the surviving cells was extracted, and the integrated sgRNA sequences were amplified by using PCR for deep sequencing. The screening revealed that SLA-DMB, TMEM239, and Rab14 exhibited the highest enrichment in the RRA scores (Fig 1H). The complete screening results are available in S5 Table.

### TMEM239 plays an important role in ASFV replication

Candidate genes identified from the GeCKO screen necessitate additional validation to eliminate false positives arising from challenges such as poor sequencing reads, copy number variations, or off-target sgRNAs [47]. SLA-DMB has been recently confirmed to be important for ASFV replication [28]. To further investigate the importance of TMEM239 and Rab14 for ASFV replication, monoclonal knockout cell lines (TMEM239$^{-/-}$ and Rab14$^{-/-}$) were generated by using the CRISPR/Cas9 gene editing system. Knockouts of TMEM239 and Rab14 in WSL cells were confirmed by using Quantitative PCR (Fig 2A and 2B) and Sanger sequencing (S2A and S2B Fig), respectively. TMEM239$^{-/-}$ and Rab14$^{-/-}$ cells were then infected with ad-7GD at an MOI of 1, and the viral titers were evaluated based on fluorescence intensity and viral DNA copies in cell supernatants at 48 hpi. Fluorescence intensity and viral DNA copies were significantly lower in TMEM239$^{-/-}$ and Rab14$^{-/-}$ cells than that in WSL cells and WSL-NC-sgRNA

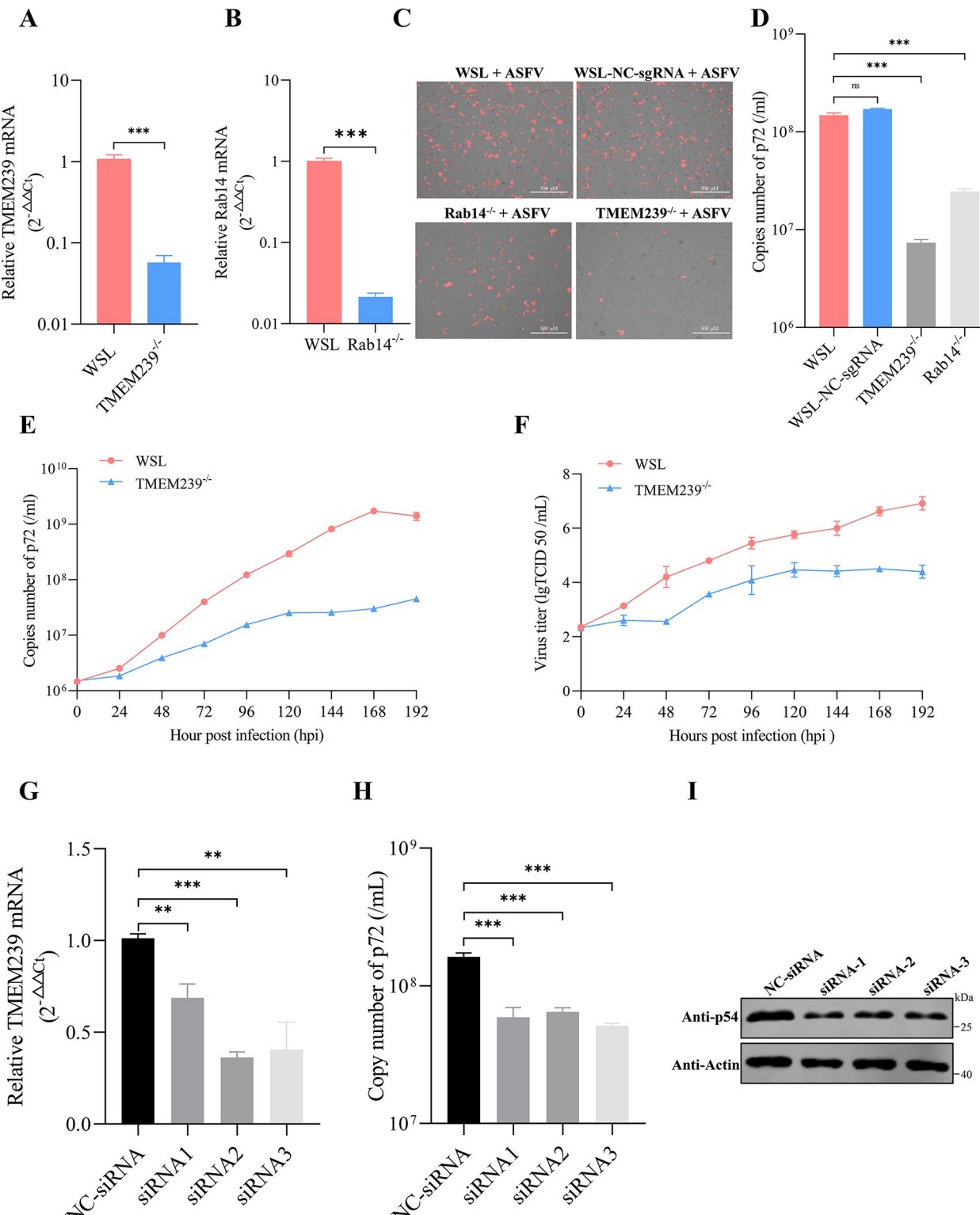

**Fig 2. TMEM239 plays an important role in ASFV replication.** (A and B) RT-qPCR analysis of TMEM239 and Rab14 transcription in TMEM239$^{-/-}$ cells and Rab14$^{-/-}$ cells. (C) Fluorescence micrographs illustrating ad-7GD replication. WSL cells, WSL cells with a non-target control sgRNA (WSL-NC-sgRNA cells), TMEM239$^{-/-}$ cells, and Rab14$^{-/-}$ cells were infected with ad-7GD (MOI = 1), and fluorescence images were collected at 48 hpi. (D) Quantification of viral genome copies (p72) in cell culture supernatants. WSL cells, WSL-NC-sgRNA cells, TMEM239$^{-/-}$ cells, and Rab14$^{-/-}$ cells were infected with ad-7GD (MOI = 1), and cell culture supernatants were collected at 48 hpi to detect viral genome copies.

***$p < 0.001$. (E and F) Multistep growth curves of ad-7GD in WSL cells and TMEM239$^{-/-}$ cells. Cells were infected with ad-7GD (MOI = 1), and samples were collected daily until the cells were destroyed by the virus. Viral replication was assessed on the basis of quantified viral genome copies (p72) and viral titers. (G) Efficiency of TMEM239 knockdown using various siRNAs. Three siRNAs specifically targeting TMEM239 were designed, synthesized, and subsequently transfected into PAMs. The mRNA level of TMEM239 was measured by RT-qPCR at 36 hpt to evaluate the knockdown efficiency. (H and I) Knockdown of TMEM239 using the three siRNAs significantly affects viral replication in PAMs. PAMs were transfected with negative control siRNA (NC-siRNA) and siRNAs specifically targeting TMEM239 for 36 h; subsequently the PAMs were infected with WT-HLJ18 (MOI = 0.1) for 48 h. *$p < 0.05$, ***$p < 0.001$.

cells (Fig 2C and 2D). Deletion of TMEM239 had a significant effect on viral replication, leading us to focus on exploring its role in ASFV replication. To exclude potential off-target effects of the sgRNA used to establish monoclonal TMEM239 knockout cells, three other sgRNAs in the GeCKO screen sgRNA library targeting the TMEM239 gene were randomly selected to generate polyclonal knockout cell lines of this gene, and these polyclonal cells were subsequently infected with ad-7GD at an MOI of 0.1. Fluorescence microscopy and viral DNA copy quantification revealed significantly reduced levels of ASFV replication in the TMEM239 knockout polyclonal cell lines compared with WSL cells at 48 hpi (S2D and S2E Fig). The multistep growth curves of ad-7GD in WSL cells and TMEM239$^{-/-}$ cells were also examined. Viral replication in WSL cells increased significantly with prolonged infection, whereas viral replication in TMEM239$^{-/-}$ cells remained at a consistently lower level (Fig 2E and 2F). To further evaluate the impact of TMEM239 on ASFV replication, the multistep growth curves of other ASFV strains including genotype I WT-SD and genotype II ad-HRB1 were measured in TMEM239$^{-/-}$ cells, and the data showed that TMEM239 deletion significantly decreased virus replication in WSL cells (S3A—S3D Fig).

PAMs serve as the primary target cells for wild-type ASFV infection. Three specific small interfering RNAs (siRNAs) targeting the porcine TMEM239 gene were synthesized and evaluated in a CCK8 assay, which showed that they had no significant cytotoxic effects on PAMs at the indicated concentrations (S2F Fig). All siRNAs exhibited significant knockdown efficiency at 36 hpt (Fig 2G). PAMs were transfected with siRNAs including a negative control siRNA (NC-siRNA), and then were infected with WT-HLJ18 (MOI = 0.1) at 36 hpt. Both cell supernatant and pellets were collected to quantify viral replication at 36 hpi. Transfection of siRNAs targeting TMEM239 resulted in a marked reduction in viral DNA copies and the expression of the viral p54 protein (Fig 2H and 2I), which suggests that TMEM239 is important for ASFV replication.

## TMEM239 is not required for ASFV adsorption and internalization

To explore which stage of the ASFV life cycle is affected by TMEM239, a virus adsorption assay was performed by using WSL cells and TMEM239$^{-/-}$ cells infected with ad-HRB1 (MOI = 5). The results showed that there was no significant difference between WSL cells and TMEM239$^{-/-}$ cells with respect to viral DNA copy number (Fig 3A). Western blot analysis confirmed that both cell types exhibited a comparable amount of viral p54 protein (Fig 3B). These results indicate that TMEM239 is not required for ASFV adsorption. Next, WSL cells and TMEM239$^{-/-}$ cells were infected with ad-HRB1 (MOI = 5) and incubated for 2 h at 4°C. An internalization assay showed that trypsin treatment effectively removed virus particles adhering to the cell surface (Fig 3C and 3D). The copy number of viral DNA and the amount of p54 protein were used to quantify the internalized virus. The data showed that the knockout of TMEM239 had no significant effect on virus internalization (Fig 3C and 3D).

## TMEM239 is important for the early step of ASFV replication

ASFV replication and assembly occur in viral factories near the cell nucleus [48]. To explore whether TMEM239 affects the formation of viral factories, WSL and TMEM239$^{-/-}$ cells were

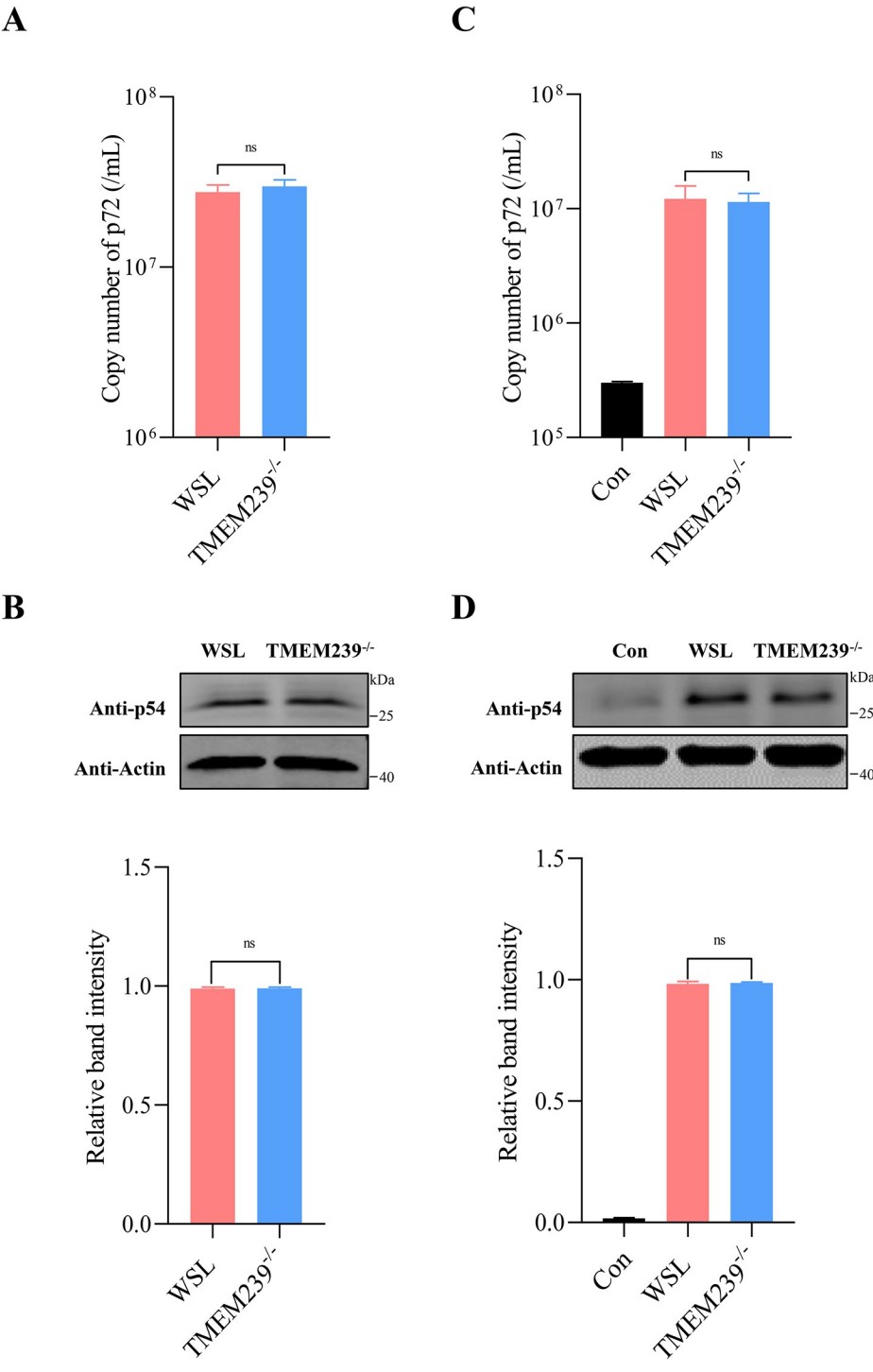

**Fig 3. TMEM239 is not required for ASFV adsorption and internalization.** (A and B) TMEM239 is not required for ASFV adsorption. WSL cells and TMEM239<sup>-/-</sup> cells were incubated with ad-HRB1 strain (MOI = 5) for 2 h at 4°C, followed by washing with ice-cold PBS to eliminate unbound virus particles. The quantification of virus particles adsorbed on the cell surface was performed using viral genome copy quantification (p72) and Western blot analysis (p54). ns: not significant. (C and D) TMEM239 is not required for ASFV internalization. WSL cells and TMEM239<sup>-/-</sup> cells were initially incubated with an ad-HRB1 isolate (MOI = 5) for 2 h at 4°C, followed by extensive washing with ice-cold PBS. Subsequently, the cells were shifted to 37°C for 2 h to facilitate the internalization of viral particles, and a trypsin treatment was applied to eliminate the remaining surface-bound virus particles. The control group (Con) underwent trypsin treatment after the 4°C incubation to determine the efficiency of removing the ASFV particles

adsorbed on the cell surface. The quantification of virus particles internalized into cells was performed using viral genome copy quantification (p72) and Western blot analysis (p54). ns: not significant.

infected with ad-HRB1 virus at an MOI of 1. At 18 hpi, the cells were fixed and stained by using an antibody against the viral capsid protein p72. DAPI staining was used to visualize nuclear and viral DNA. WSL cells displayed clear fluorescence of p72 and viral factory formations after ASFV infection, whereas fluorescence of p72 was too weak in TMEM239$^{-/-}$ cells (Fig 4A). These results indicate that knockout of TMEM239 significantly affects viral factory formation. The formation of viral factories is modulated by various ASFV viral proteins. We therefore evaluated the effect of TMEM239 knockout on ASFV gene expression, including the early representative gene *CP204L* (encoding p30 protein) and the late representative gene *B646L* (encoding p72 protein). Relative to 0 hpi, the *CP204L* and *B646L* mRNA transcript levels in ASFV-infected WSL cells were significantly increased at 2 hpi and 8 hpi, respectively. In contrast, the *CP204L* and *B646L* mRNA transcript levels in TMEM239$^{-/-}$ cells remained low at these timepoints (Fig 4B and 4C). Western blot analysis validated these findings (Fig 4D). ASFV-infected WSL cells showed a significant increase in the expression of p30 and p72 protein at 4 hpi and 8 hpi, respectively. However, the expression of p30 and p72 in TMEM239$^{-/-}$ cells was significantly lower than that in WSL cells at each corresponding timepoint (Fig 4E and 4F). Taken together, the results demonstrate the importance of TMEM239 in the early stage of ASFV replication.

## TMEM239 knockout impedes ASFV entry into early endosomes

To further investigate the role of TMEM239 in ASFV replication, we identified host and viral proteins that interact with TMEM239 by using a pulldown assay and LC-MS analysis. Flag-tagged TMEM239 plasmid and control plasmid (pCAGGS-Flag) were transfected into WSL cells, respectively, and ad-HRB1 was inoculated at MOI = 5 into one of the groups of cells transfected with TMEM239 plasmid. All cell lysates were immunoprecipitated using an anti-Flag antibody for LC-MS analysis. Immunoprecipitated samples from cells transfected with the control plasmid were used as negative controls to eliminate nonspecific interactions. In addition, cell lysates were immunoprecipitated using an anti-Flag antibody and stained with silver to visualize TMEM239 and its binding proteins (Fig 5A). A total of 62 cellular proteins that specifically interacted with TMEM239 were identified in the uninfected virus group (S6 Table). In the infected virus group, only 35 host proteins specifically interacted with TMEM239, with no viral proteins being identified (S7 Table). GO enrichment analysis of these proteins that interacted with TMEM239 in the uninfected virus group was performed to predict their biological functions by using the OmicsBean analysis tool (Fig 5B). The findings suggested that host proteins interacting with TMEM239 are involved in small GTPase-mediated signal transduction and transmembrane transport. KEGG pathway analysis showed that host proteins that interact with TMEM239 are highly associated with pathways including "Transport and catabolism", "signaling molecules and interaction", "signal transduction", and "membrane transport" (Fig 5C).

Once ASFV is internalized into cells, its early intracellular transport plays a crucial role in the virus replication process [49]. LC-MS analysis unveiled the interaction of TMEM239 with the early endosomal marker Rab5A (S6 and S7 Tables). To further examine the interaction between TMEM239 and Rab5A, we performed co-immunoprecipitation (Co-IP) and confocal experiments. HEK-293T cells were transfected with plasmids expressing Flag-tagged TMEM239 (pTMEM239-Flag) and Myc-tagged Rab5A (pRab5A-Myc) together or individually. TMEM239-Flag was eluted and co-immunoprecipitated with Myc-tagged Rab5A protein

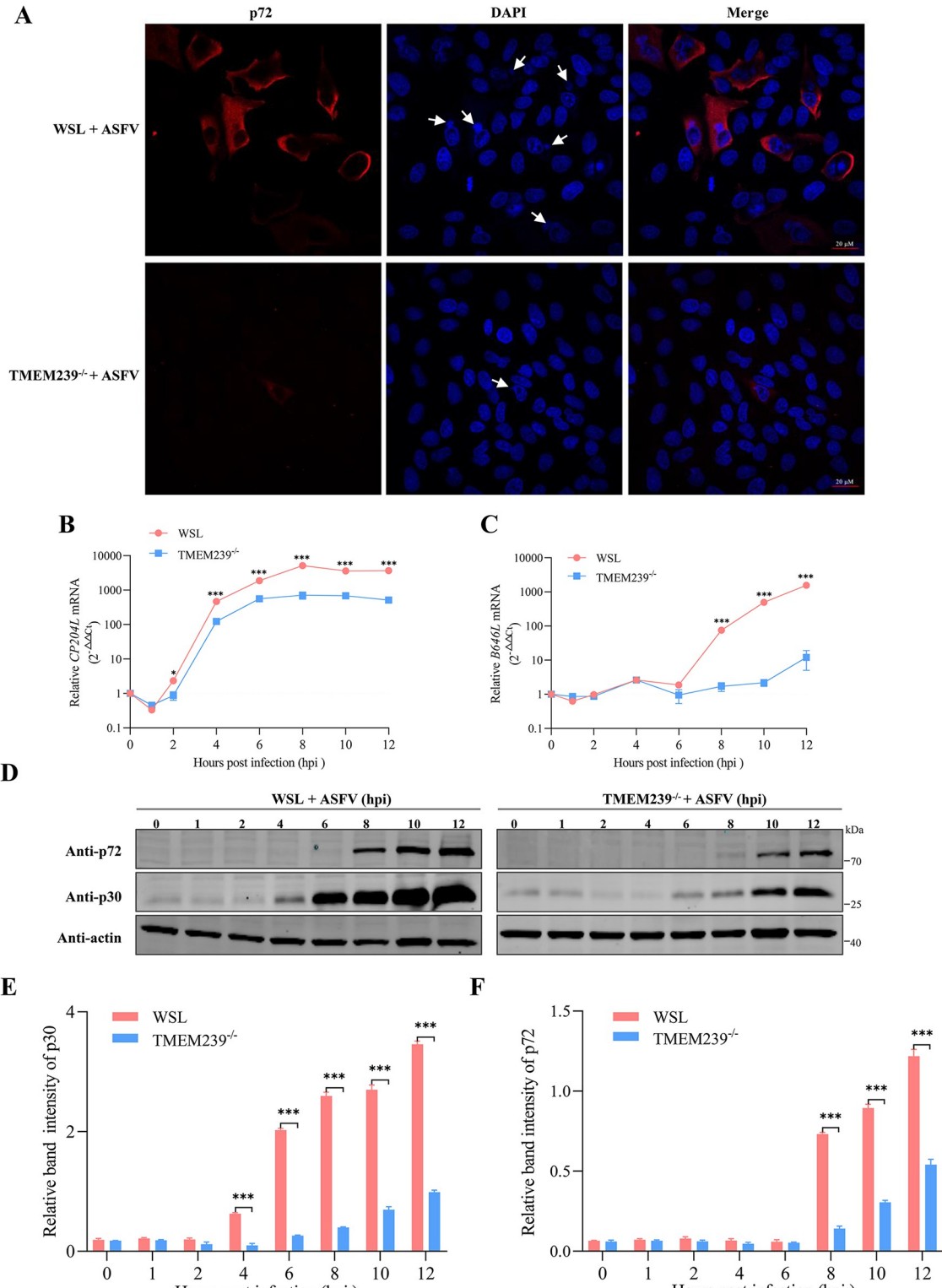

**Fig 4. TMEM239 is important for the early step of ASFV replication.** (A) Knockout of TMEM239 affects the formation of viral factories. WSL cells and TMEM239$^{-/-}$ cells were infected with ad-HRB1 isolate (MOI = 1) for 18 h, followed by fixation with methanol. Subsequent immunofluorescence analysis targeted p72 (red), whereas nuclear and viral DNA was visualized with DAPI staining (blue). The arrows indicate virus factories. (B and C) Knockout of TMEM239 affects the transcription of the viral early gene *CP204L* (p30) and late gene *B646L* (p72). WSL cells and TMEM239$^{-/-}$ cells were incubated with ad-HRB1 isolate (MOI = 5) for 2 h at

4˚C, followed by extensive washing with ice-cold PBS. Then, the cells were shifted to 37˚C and collected at multiple timepoints (0 h, 1 h, 2 h, 4 h, 6 h, 8 h, 10 h, and 12 h). Total RNA was extracted to detect the transcription of *CP204L* and *B646L*. $^*p < 0.05$, $^{***}p < 0.001$. (D) Knockout of TMEM239 affects the translation of the viral early gene *CP204L* (p30) and the late gene *B646L* (p72). WSL cells and TMEM239$^{-/-}$ cells were incubated with ad-HRB1 isolate (MOI = 5) for 2 h at 4˚C, followed by extensive washing with ice-cold PBS. Then, the cells were shifted to 37˚C and collected at multiple timepoints (0 h, 1 h, 2 h, 4 h, 6 h, 8 h, 10 h, and 12h). Total protein was extracted to detect the expression of viral genes p30 and p72. (E and F) Quantitative analysis of the expression of the viral proteins (p30 and p72) at different timepoints in WSL cells and TMEM239$^{-/-}$ cells. $^{***}p < 0.001$.

when they were co-expressed (Fig 6A). Further, the co-localization of TMEM239 and Rab5A in HEK-293T cells co-transfected with plasmids overexpressing these proteins was evaluated by using confocal microscopy. Notably, a robust co-localization signal between TMEM239 and Rab5A was detected in cells (Fig 6B). These results indicate that TMEM239 interacts with Rab5A in cells. Shortly after infection, the ASFV virions enter into cells via clathrin-mediated endocytosis (CME) and the capsid protein p72 can be detected in early endosomes colocalizing with specific markers such as EEA1 and Rab5A [17]. Dynasore, a specific inhibitor of the GTPase activity of dynamin, was used to explore ASFV entry into WSL cells. The results demonstrate that dynasore inhibits ASFV internalization (Fig 6C and 6D), suggesting that ASFV virions enter the WSL cells via CME. Besides, the Co-IP assays of TMEM239 and eight viral proteins revealed that pE248R involved in early postentry event of ASFV [50] was a TMEM239 binding partner (Fig 6E). We then wondered whether TMEM239 affects ASFV virions entry into early endosomes. WSL and TMEM239$^{-/-}$ cells were incubated with ad-HRB1 (MOI = 20) for 2 h at 4˚C, and were then washed with ice-cold PBS and incubated at 37˚C for 30 minutes, 45 minutes, and 60 minutes, respectively. These cells were then fixed and double stained with antibodies against p72 and Rab5A. In addition, a negative control group incubated at 37˚C for 45 minutes was set up to explore the colocalization of p72 with the late endosomal marker Rab7 (S4A Fig). Confocal microscopy analysis showed that p72 and Rab5A were mainly colocalized in WSL cells (Fig 7A) but not in TMEM239$^{-/-}$ cells at these three timepoints (Fig 7B). Quantitative analysis of the p72 and Rab5A signals showed a higher Pearson's correlation in WSL cells than that in TMEM239$^{-/-}$ cells (Fig 7C). Transferrin serves as a marker to monitor internalization through CME [51]. Cy3-labeled transferrin was used to assess the endosomal transport function in TMEM239$^{-/-}$ cells. The results showed that TMEM239$^{-/-}$ cells exhibited the efficient capacity of transferrin transport comparable to that of wild-type WSL cells, indicating that TMEM239 knockout does not interfere with the normal function of endosomes (S4B Fig). These results indicate that TMEM239 knockout specifically impedes ASFV entry into early endosomes.

## ASFVs exhibit lower replication in PBMCs from TMEM239$^{-/-}$ piglets

Given that TMEM239 is involved in ASFV replication *in vitro*, we next investigated whether *in vivo* deletion of TMEM239 could similarly impede ASFV replication. Three sgRNAs were designed each upstream (~490 bp) and downstream (~6100 bp) of the TMEM239 gene coding sequence, and the gene editing efficiency of these sgRNAs was evaluated by using a T7E1 assay. The sgRNAs showing the most significant gene editing efficiency (S2 Table) were then introduced into porcine embryonic fibroblast (PEF) cells via electroporation together with Cas9 protein to completely knock out the coding region of the TMEM239 gene. Through PCR characterization (S5A Fig) and Sanger sequencing verification (Fig 8A), a monoclonal PEF cell clone (#1) with complete deletion of the TMEM239 coding region was successfully obtained and used as a donor for SCNT (Somatic Cell Nuclear Transfer) to generate reconstructed embryos, which were transplanted into two surrogates (#355 and #393). Consequently, nine piglets were born, with 393–5 succumbing shortly after birth. Further genotyping analysis

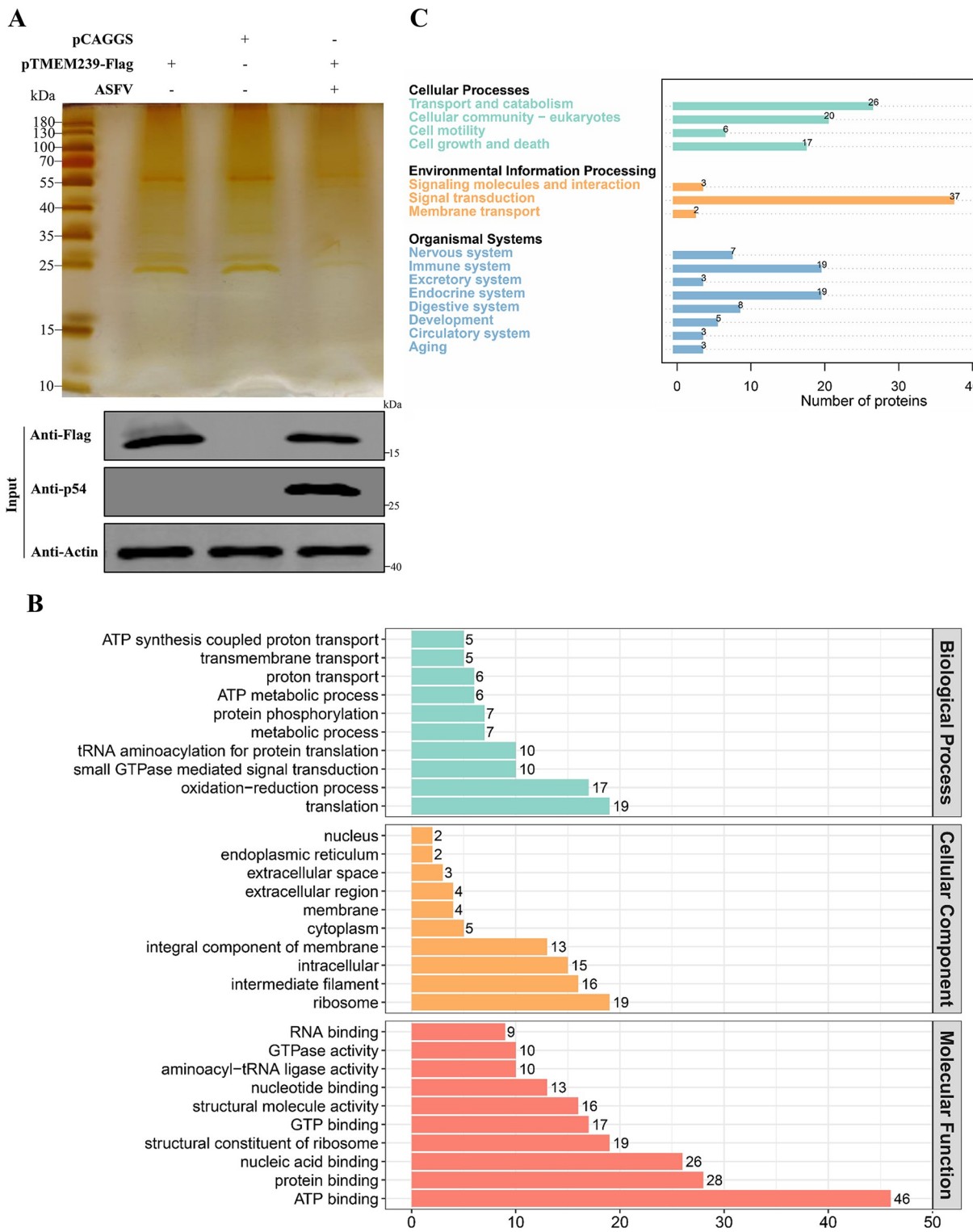

**Fig 5. Proteomic analysis of host proteins interacting with TMEM239.** (A) Detection of exogenous TMEM239 expression in WSL cells and visualization of proteins that interact with TMEM239 by using silver staining. pTMEM239-Flag (6μg) and empty pCAGGS-Flag (6μg) were transfected into WSL cells, respectively. At 24 hpt, ad-HRB1 was inoculated at MOI = 5 into one of the groups of cells transfected with TMEM239 plasmid. Anti-Flag (M2) agarose beads were used to pull down TMEM239 and its binding proteins 24 h after transfection. (B) Gene ontology (GO) analysis based on the interaction between TMEM239 and its binding proteins in the uninfected virus group. The GO distributions of all

proteins were divided into three types (biological processes, composition category, and function category). (C) KEGG pathway enrichment analysis based on the interaction between TMEM239 and its binding proteins in the uninfected virus group. Cellular processes, Environmental information processing, and organismal systems are listed.

revealed that 355–4 was a heterozygous knockout (Figs 8B and S5B). All TMEM239$^{-/-}$ knock-out piglets were in good health after birth. According to the normal immunization program, these piglets were vaccinated with vaccinations including swine fever virus vaccine, porcine reproductive and respiratory syndrome virus (PRRSV) vaccine, and porcine circovirus vaccine. However, during subsequent feeding, one of the piglets (#355–2) displayed symptoms such as diarrhea, swollen joints, and lethargy. It was further identified that the piglet was infected with PRRSV. Given the limited number of the piglets, we isolated PBMCs from the TMEM239 homozygous knockout (TMEM239$^{-/-}$) piglets to investigate the effect on ASFV replication *ex vivo*. The results of flow cytometry analysis revealed no significant differences in the proportions of CD45$^-$, CD45$^+$, CD3$^+$, CD4$^-$CD8$^+$, CD4$^+$CD8$^-$, CD4$^+$CD8$^+$, CD4$^-$CD8$^-$, B cell, and monocyte populations between TMEM239$^{-/-}$ pigs and wild-type pigs PBMCs (S5C—S5E Fig). Wild-type piglet PBMCs (WT PBMCs) and TMEM239$^{-/-}$ piglet PBMCs (TMEM239$^{-/-}$ PBMCs) were infected with ad-7GD at an MOI of 1, followed by collection of fluorescence images and supernatants at 72 hpi. In comparison to WT PBMCs, the TMEM239$^{-/-}$ PBMCs displayed significantly lower fluorescence intensity (S6A Fig) and lower viral DNA copy numbers (Fig 8C). Furthermore, flow cytometry analysis of ASFV-infected PBMCs showed that monocytes were the main infected cells in the WT PBMCs and TMEM239$^{-/-}$ PBMCs (S6B and S6D Fig). Lastly, the WT-HLJ18 strain was used to infect WT PBMCs and TMEM239$^{-/-}$ PBMCs (MOI of 0.1), and cell supernatants were harvested for viral titration at 72 hpi. Quantification of viral DNA copies (Fig 8D) and HAD$_{50}$ (Fig 8E) showed significantly reduced replication of WT-HLJ18 in TMEM239$^{-/-}$ PBMCs. These results indicate that ASFV replication was significantly suppressed in TMEM239$^{-/-}$ PBMCs.

## Discussion

ASFV is an intracellular parasitic pathogen that is heavily reliant on host factors for its life cycle [52]. In this study, we developed a GeCKO screen resource and identified TMEM239 as a potential host factor required for ASFV replication in the WSL cell line. Knock-out of TMEM239 in WSL cells significantly suppressed ASFV replication. Further studies showed that TMEM239 interacts with the early endosome marker Rab5A, and plays an important role in the early stages of ASFV infection. Upon TMEM239 knockout, virus particles were internalized into the cell, and the colocalization of the ASFV capsid protein p72 and Rab5A was markedly disrupted. These findings indicate that TMEM239 is required for a post-endocytic stage of ASFV entry. In addition, we generated TMEM239 knockout piglets using SCNT technology and showed that ASFV replication is significantly reduced in TMEM239$^{-/-}$ PBMCs.

The GeCKO screen strategy is a potent method for identifying host factors essential for viral infection [53,54]. To create this screening resource, a WSL cell line stably expressing Cas9 with high nuclease cleavage activity was developed (Fig 1A–1D). Additionally, we designed and synthesized 186,510 specific sgRNAs targeting 20,580 porcine genes using genome assembly Sus scrofa 11.1. Compared to other porcine CRISPR libraries, such as SsCRISPRko.v1 [55], the sgRNA library for CRISPR knockout developed in this study encompasses a larger number of sgRNAs and more sgRNAs targeting each gene. Deep sequencing revealed that while 4,795 sgRNAs were absent from the CRISPR knockout plasmid library we constructed, only 23 genes were lacking. Low-stringency screening may increase the background during screening due to the presence of uninfected cells [56]. The current study used a high-stringency

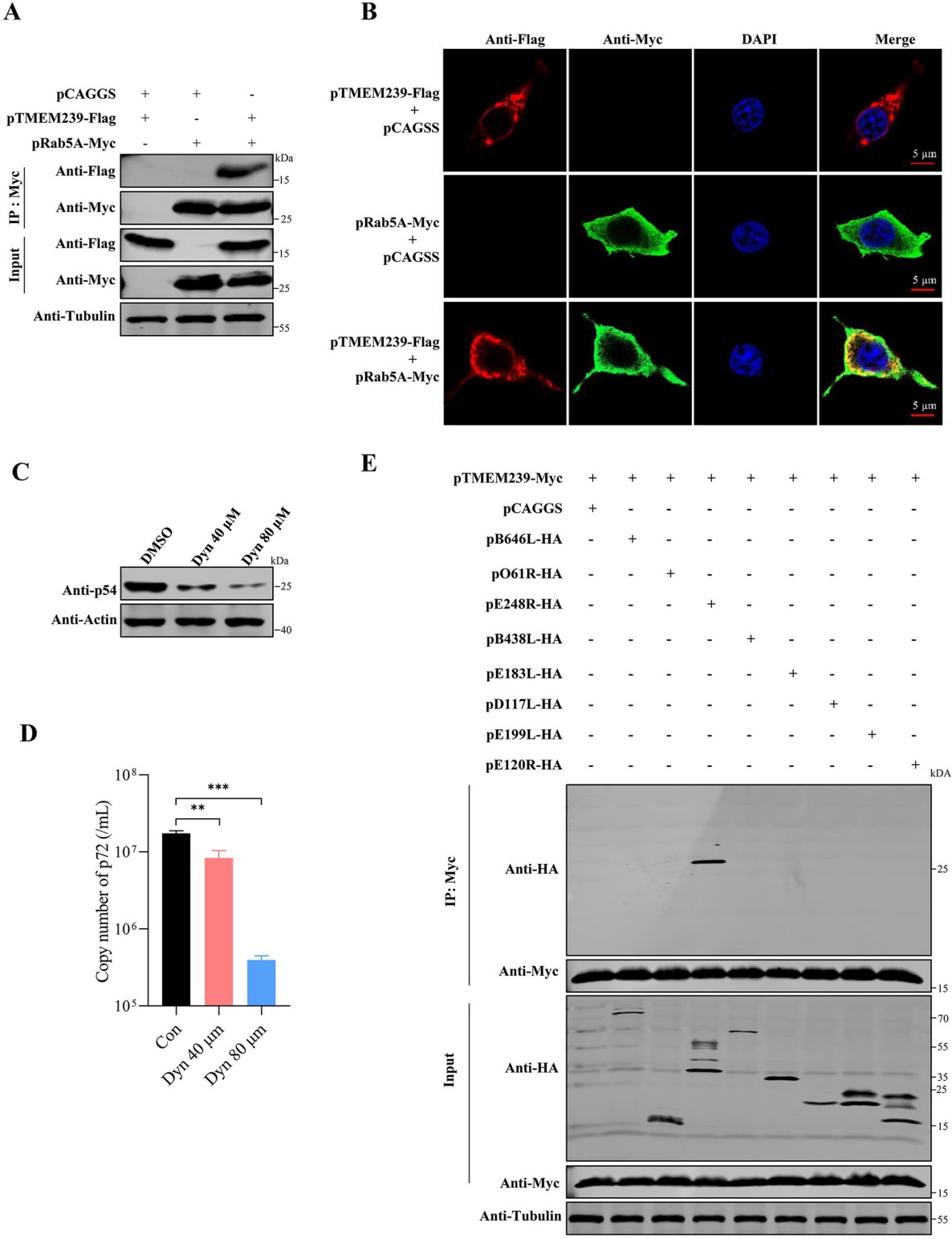

**Fig 6. TMEM239 interacts with Rab5A and the virus protein pE248R.** (A) TMEM239 and Rab5A co-immunoprecipitation (Co-IP) assay. HEK-293T cells were transfected with pTMEM239-Flag and pRab5A-Myc either together or separately. At 36 hpt, cell lysates were collected for co-immunoprecipitation with beads conjugated with Myc antibody. Tubulin was used as an internal loading control. (B) Colocalization analysis of TMEM239 and Rab5A. Colocalization of TMEM239 and Rab5A was assessed in HEK-293T cells co-transfected with pTMEM239-Flag and pRab5A-Myc by use of confocal microscopy. Immunofluorescence analysis was performed against Flag (red) and

Myc (Green). Nuclear DNA was visualized by DAPI staining (blue). (C and D) Dynasore inhibits ASFV internalization into WSL cells. WSL cells pre-treated for 15 minutes with the inhibitor for clathrin-dependent endocytosis dynasore were incubated with ad-HRB1 isolate (MOI = 5) for 2 h at 4°C, followed by extensive washing with ice-cold PBS. Subsequently, the cells were shifted to 37°C for 30 min to facilitate the internalization of viral particles, and a trypsin treatment was applied to eliminate the remaining surface-bound virus particles. Western blot analysis (p54) and quantification of viral genome copies (p72) were used to quantify virus particles internalized into cells. ***$p < 0.001$. (E) TMEM239 and viral proteins co-immunoprecipitation (Co-IP) assay. HEK-293T cells were transfected with pTMEM239-Myc and HA-tagged viral protein expression plasmids either together or separately. At 36 hpt, cell lysates were collected for co-immunoprecipitation with beads conjugated with Myc antibody. Tubulin was used as an internal loading control.

screening strategy in which surviving cells from the ASFV-infected WSL-GeCKO cell library were re-infected with the virus for two additional rounds of screening to identify the host factors required for ASFV replication. Deep sequencing data from the screening were analyzed using the MAGeCK algorithm, and three host factors (SLA-DMB, TMEM239, and Rab14) had RRA values less than $10^{-4}$. SLA-DMB has previously been identified as an essential host factor for ASFV replication [28]. Our study further demonstrated that ASFV replication was significantly reduced in both TMEM239 and Rab14 knockout cell lines compared to WSL cells. Thus, the GeCKO screen resource and the high-stringency screening strategy developed in this study are sound.

In the present study, Rab14 was identified as an important host factor for ASFV replication. Rab14, a member of the RAS oncogene family, is involved in intracellular membrane trafficking [57], the endocytic pathway of cationic substances [58], and switching between GTP/GDP [59]. As a host factor required for viral infection, Rab14 has also been shown to be involved in the replication of viruses such as human immunodeficiency virus (HIV) [60], Ebola virus (EBOV) [61], and classical swine fever virus (CSFV) [62]. However, there are fewer studies of TMEM239, a member of the transmembrane (TMEM) protein family. The TMEM protein family comprises numerous members that traverse the lipid bilayer and are integral components of various biological membranes, including the endoplasmic reticulum, lysosomes, mitochondria, and the Golgi apparatus [63]. TMEM proteins are widely expressed in diverse tissues and are believed to act as channels facilitating the transport of various substances [64]. Recently, numerous studies have elucidated the pivotal role of TMEM proteins in viral infection. TMEM106B was shown to be a receptor mediating ACE2-independent SARS-CoV-2 cell entry [56]. Multiple independent GeCKO screens revealed that TMEM41B is a key host factor required for different coronaviruses including SARS-CoV-2 [65], transmissible gastroenteritis virus (TGEV), and porcine delta-corona virus (PDCoV) [66]. TMEM251, also known as LYSET, is essential for SARS-CoV-2 infection [67].

Limited research on TMEM239 prompted us to perform a proteomics-based gene function prediction. The GO and KEGG enrichment analysis based on the proteins that interact with TMEM239 showed that TMEM239 is involved in GTPase-mediated signal transduction and transmembrane transport. Several studies have demonstrated that the primary internalization mechanisms of ASFV into host cells involve two pathways: CME (Clathrin-mediated Endocytosis) and macropinocytosis [14,49]. Regardless of the internalization pathway used, the incoming viruses are transported into the endolysosomal pathway via multivesicular endosomes. We performed the GeCKO screening in WSL cells. A previous study revealed that ASFV virions are internalized into WSL cells via CME [13]. During CME, virions are internalized with their specific receptors via coated membrane invaginations and rapidly form coated vesicles [68]. Clathrin, a major component of the invaginations and vesicles, is recruited to the inner side of the plasma membrane with the participation of other accessory molecules such as AP180, Eps15, amphoterin, and endophylline, and its interaction with the AP-2 adapter complex forms coated vesicles to endocytose the virus particles into the cell [69]. Subsequently, the

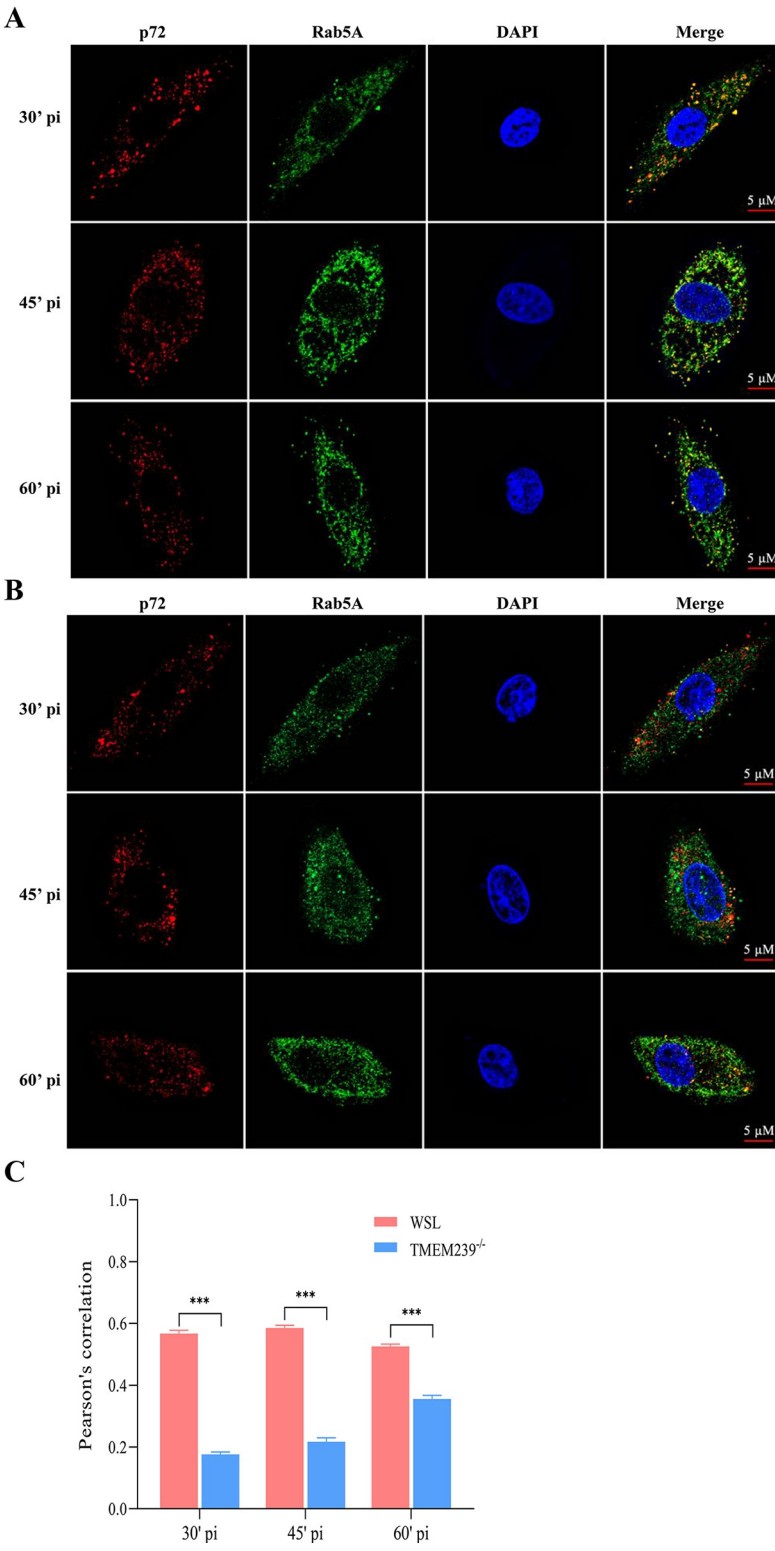

**Fig 7. Knockout of TMEM239 affects colocalization of the viral p72 protein with Rab5A.** (A and B) Knockout of TMEM239 affects the colocalization of the viral p72 protein with Rab5A. WSL cells (A) and TMEM239$^{-/-}$ cells (B) were incubated with ad-HRB1 isolates (MOI = 20) for 2 h at 4°C and then washed with ice-cold PBS to remove unbound virus particles. Then, ice-cold medium was added, and the cells were incubated at 37°C for 30, 45, and 60 minutes before being fixed in methanol. Immunofluorescence analysis was performed against p72 (red) and Rab5A (Green).

Nuclear DNA was visualized by DAPI staining (blue). (C) Quantification of p72-Rab5A Co-localization. p72-Rab5A co-localization was quantified by using Pearson's correlation coefficient method. Cells co-stained with p72 and Rab5A antibodies were randomly selected for co-localization analysis by using ImageJ software. ***$p < 0.001$.

vesicle is separated from the plasma membrane through the action of the GTPase dynamin [70]. Ultimately, the vesicle sheds clathrin and matures into early endosomes. When ASFV virion enters the endosomal transport system, the interaction of host factor Niemann-Pick C type 1 (NPC1) with the viral protein pE248R facilitates virion transport within the endosomal network [71]. In our study, we also discovered the interaction between TMEM239 and pE248R. Besides, LC-MS analysis and subsequent experiments confirmed that TMEM239 interacts with Rab5A and localizes in early endosomes. Taking all of these findings together, we propose that TMEM239 is involved in ASFV entry into early endosomes. Confocal microscopy assays were employed to investigate the ASFV virions transported into early endosomes shortly after infection. The results showed that the level of colocalization of p72 with Rab5A in TMEM239$^{-/-}$ cells is lower than that in WSL cells and transferrin uptake assay revealed that TMEM239 knockout did not interfere with the normal function of endosomes, which suggests that TMEM239 knockout specifically impedes ASFV entry into early endosomes.

Although TMEM239 knockout affects ASFV replication, we observed that different ASFV strains replicate at different levels in TMEM239$^{-/-}$ cells. Specifically, the TCID$_{50}$ assay showed a 2-log reduction of ad-7GD replication in TMEM239$^{-/-}$ WSL cells, a reduction of more than 1-log for ad-HRB1 replication, and less than a 1-log reduction for WT-SD replication. This variability is likely attributable to viral genome differences of ASFVs used in this study. The genomic identity between genotype II ad-7GD and ad-HRB1 was 99.56%, while genotype I WT-SD showed 97.25% and 97.48% genomic identity with ad-7GD and ad-HRB1, respectively. Therefore, the effect of host factors on virus replication may be associated with virus variation. CD163, the primary receptor for porcine reproductive and respiratory syndrome virus (PRRSV) cell entry, exhibits genotype-specific interactions; pigs with specific knockout of exon 5 of CD163 are resistant to genotype I PRRSV but not genotype II PRRSV [43]. Given the huge genome of ASFV, with more than 170 Kb, and its intricate viral variation, identifying a host factor universally involved in the replication of various genotypes of viruses remains challenging. There is no any cell line that could support efficient replication of different wild-type ASFVs [72], so we used the gene-deleted and cell-adapted ASFVs to investigate important host factors associated with ASFV replication by using the GeCKO screen in WSL cells. It is noteworthy that the GeCKO screen in WSL cells may not provide an ideal representation of wild-type ASFV replication in its susceptible PAM. Therefore, we knocked down TMEM239 expression by using siRNA to further evaluate its impact on the replication of WT-HLJ18 strain in PAM. Additionally, an interesting discovery in the LC-MS analysis is the interaction between TMEM239 and Rab14. Given the known functions of the two genes, it is conceivable that they cooperatively regulate biological processes encompassing membrane transport and energy conversion.

As an important source of meat, domestic pigs rank among the most consumed livestock globally. However, various infectious pathogens pose a great challenge to the pig industry. Genome editing presents novel opportunities to improve livestock breeding with a focus on disease resistance, enabling the direct translation of laboratory research findings into the production of disease-resistant animals. Porcine reproductive and respiratory syndrome (PRRS) is one of the diseases that threatens the pig industry, and CD163 is a receptor that facilitates PRRSV entry into target cells [73]. CD163-deficient pigs generated by genome editing are significantly resistant to PRRSV infection [74]. Similarly, transmissible gastroenteritis virus

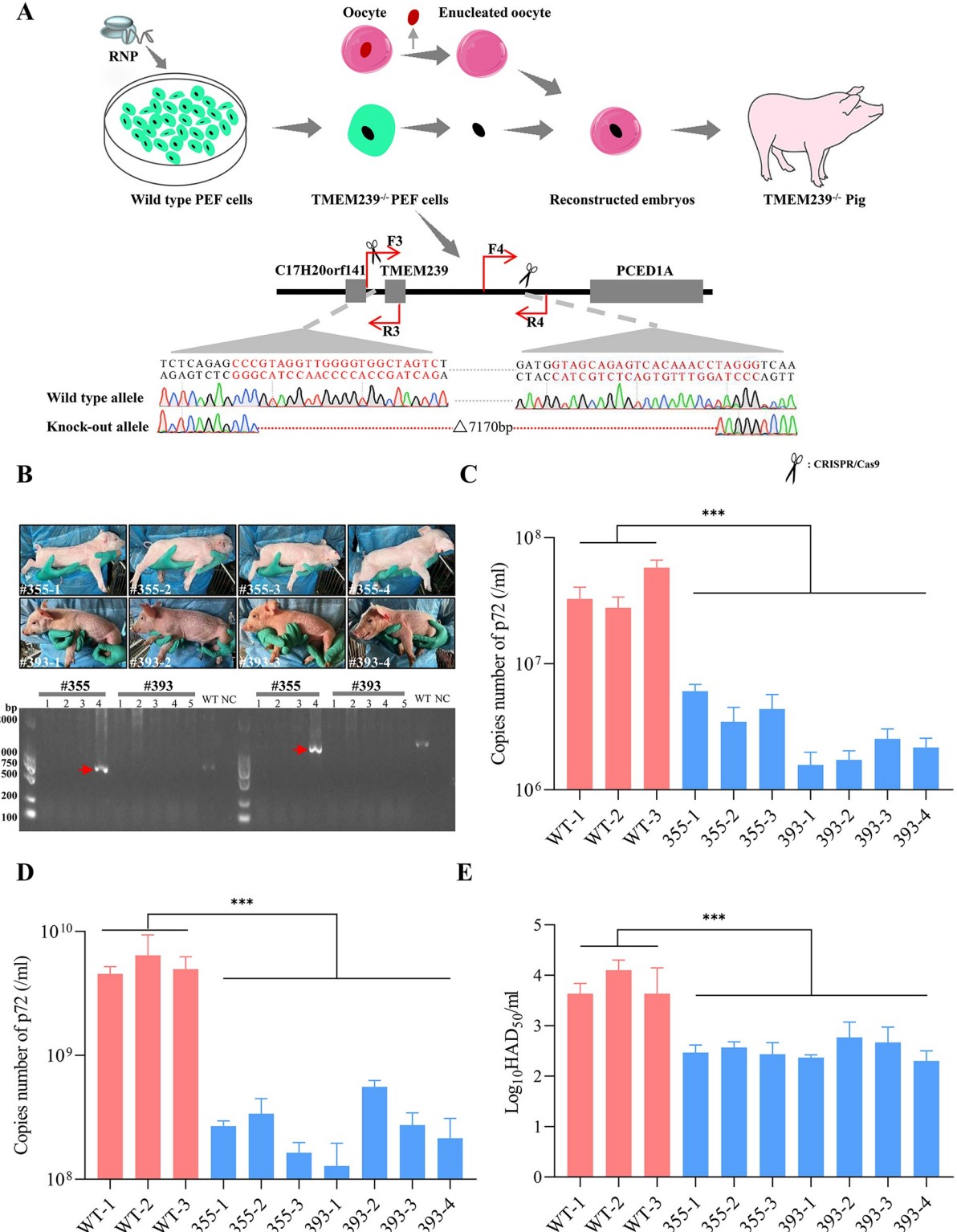

**Fig 8. ASFVs exhibit lower replication in TMEM239⁻ᐟ⁻ PBMCs.** (A) SCNT schematic for TMEM239 knockout piglets. sgRNAs and Cas9 protein (RNP) were co-electroporated into PEF cells, generating cells that completely lacked the TMEM239 coding region. Then, SCNT technology was used to generate TMEM239 knockout piglets. F3/R3, F4/R4, and F3/R4 are primer pairs used to identify the genotype of the SCNT-generated piglets. (B) Genotype identification of SCNT-producing Piglets. DNA extracted from the ear margin of the piglets was subjected to PCR identification. F1/R1 and F2/R2 primers were used to ascertain the genotypes of the piglets. PCR amplification products of

wild-type piglets were 615 bp (left panel) and 930 bp (right panels) bands, respectively, whereas the PCR amplification results for the TMEM239 knockout piglets were negative. The red arrows indicate the wild-type bands from piglet #355–4. WT, Wild-type piglet, NC, Negative control. (C) Quantification of ad-7GD Strain Viral Genome Copies (p72). WT PBMCs and TMEM239$^{-/-}$ PBMCs were individually infected with the ad-7GD strain at an MOI of 1. Supernatants were collected at 72 hpi, and viral genome copies (p72) were quantified. (D and E) Quantification of WT-HLJ18 strain viral genome copies (p72) and HAD$_{50}$. WT PBMCs and TMEM239$^{-/-}$ PBMCs were infected with the WT-HLJ18 strain at an MOI of 0.1. At 72 hpi, supernatants were collected, and viral genome copies, as well as the HAD$_{50}$ values, were quantified. WT-1, WT-2, and WT-3 indicate the numbers of wild-type piglets selected for the current experiment. Piglet numbers 355-(1–4) and 393-(1–5) correspond to piglets born to surrogate sows numbered #355 and #393.

(TGEV) causes severe diarrhea and high mortality in pre-weaned piglets, but pigs lacking aminopeptidase N, a receptor for TGEV, generated by genome editing are resistant to TGEV infection. The current *in vitro* findings reveal the potential of TMEM239 as a candidate gene for breeding ASFV-resistant pigs [75]. To further investigate whether deletion of TMEM239 *in vivo* similarly hinders ASFV replication, we used SCNT technology to generate TMEM239 knockout pigs. PCR identification confirmed the successful acquisition of seven TMEM239 homozygous knockout piglets, indicating that TMEM239 is not an embryonic lethal gene. Given the limited number the piglets, we used PBMCs from the TMEM239 homozygous knockout piglets to investigate the extent of virus replication. Subsequent virus infection experiments confirmed that virus replication in TMEM239$^{-/-}$ PBMCs was significantly reduced compared with that in WT-PBMCs, whether inoculated with the ad-7GD strain or the WT-HLJ18 strain. Given that numerous TMEM239$^{-/-}$ PBMCs remained infected, we speculated that PBMCs include different types of ASFV-susceptible cells, such as monocytes, macrophages, and granulocytes, which may be at different stages of differentiation [76,77]. Perhaps TMEM239 supports ASFV replication in different kinds of cells with a cell-type dependent manner.

In conclusion, we successfully developed a GeCKO screening resource and identified TMEM239 as an important host factor for ASFV replication. We further demonstrated that TMEM239 interacts with Rab5A and plays an important role in ASFV entry into early endosomes. Our findings provide insights for the development of ASF-resistant breeding.

## Supporting information

**S1 Fig. Viral replication kinetics of ASFVs used in this study.** (A and B) Multistep growth curves of WSL-adapted strains (ad-7GD and ad-HRB1) and wild-type strains (WT-SD and WT-HLJ18) in PAMs. PAMs were infected with ad-HRB1, WT-SD, and WT-HLJ18 at a Multiplicity of Infection (MOI) of 0.1, while ad-7GD was infected at an MOI of 1. Samples were collected daily until cell destruction by virus infection. Viral replication was characterized by quantifying viral genome copies (p72) and viral titers. (C) Replication of ad-7GD in WSL and WSL-Cas9-sgRNA cells. Fluorescence micrographs depicting the replication of ad-7GD in WSL cells and WSL-Cas9-sgRNA cells following incubation with the virus (MOI = 1) for 48 h. (D) Quantification of viral genome copies (p72) in the cell culture supernatants of WSL cells and WSL-Cas9-sgRNA cells. ns: not significant.
(TIF)

**S2 Fig. The TMEM239 gene plays an important role in ASFV replication.** (A) Identification of TMEM239 gene monoclonal knockout cell lines through Sanger sequencing. (B) Identification of Rab14 gene monoclonal knockout cell lines through Sanger sequencing. (C) RT-qPCR analysis of TMEM239 transcription in polyclonal knockout cell lines. ***p < 0.001. (D) Replication of ad-7GD in TMEM239 polyclonal knockout cell lines. Fluorescence micrographs illustrating the replication of ad-7GD in WSL cells and various TMEM239 polyclonal knockout cell lines generated through the transfection of different sgRNAs. All cells were incubated

with virus (MOI = 1), and fluorescence micrographs were captured at 48 hpi. (E) Quantification of viral genome copies (p72) in the cell culture supernatants of WSL cells and diverse TMEM239 polyclonal knockout cell lines. ns: not significant. ***p < 0.001. (F) Assessment of cytotoxic effects of siRNAs on PAMs. PAMs were individually transfected with all siRNAs and incubated for 24 h. Then, the influence of these siRNAs on cell viability was assessed using the CCK8 assay.
(TIF)

**S3 Fig. Multistep growth curves of ad-HRB1 and WT-SD in WSL cells and TMEM239-/- cells.** (A and B) Multistep growth curves of ad-HRB1 in WSL cells and TMEM239$^{-/-}$ cells. Cells were infected with ASFV Genotype II ad-HRB1 strain (MOI = 1), and samples were collected daily until cell destruction by the virus. Viral replication was characterized by quantifying viral genome copies (p72) and viral titers. (C and D) Multistep growth curves of WT-SD in WSL cells and TMEM239$^{-/-}$ cells. Cells were infected with ASFV Genotype I WT-SD strain (MOI = 1), and samples were collected daily until cell destruction by the virus. Viral replication was characterized by quantifying viral genome copies (p72) and viral titers.
(TIF)

**S4 Fig. Knockout of TMEM239 does not affect transferrin entry into early endosomes.** (A) Colocalization analysis of viral p72 and Rab7 was performed at 45 minutes after viral infection. WSL cells and TMEM239$^{-/-}$ cells were incubated with ad-HRB1 isolates (MOI = 20) for 2 h at 4˚C and then washed with ice-cold PBS to remove unbound virus particles. Then, ice-cold medium was added, and the cells were incubated at 37˚C for 45 minutes before being fixed in methanol. Immunofluorescence analysis was performed against p72 (red) and Rab7 (green). Nuclear DNA was visualized by DAPI staining (blue). (B) Transferrin uptake assay in WSL cells. After serum starvation for 30 minutes, WSL cells were incubated with 50 μg/mL Cy3-labelled transferrin in DMEM for 20 minutes at 4˚C for binding. Subsequently, cells were washed with ice-cold PBS to remove unbound protein, shifted to 37˚C for 20 minutes to facilitate transferrin internalization, and subjected to trypsin treatment to eliminate remaining surface-bound protein before fixation in methanol. Immunofluorescence analysis was performed against transferrin (red) and Rab5A (green). Nuclear DNA was visualized by DAPI staining (blue). The arrows indicate transferrin transferred to early endosomes.
(TIF)

**S5 Fig. Genotype identification of PEF cells with TMEM239 gene monoclonal knockout and phenotype of SCNT-producing piglets.** (A) Identification of TMEM239 gene monoclonal knockout PEF cell clones through PCR. Primer pair F3/R4 was used to identify PEF cell clones generated by the CRISPR/Cas9 gene editing system with complete deletion of the coding region of the TMEM239 gene. Specifically, primer pair F3/R4 amplifies a 7678-bp band in wild-type PEF cells, whereas in PEF cells, in which the coding region of the TMEM239 gene is entirely deleted, a ~513-bp band is amplified. Clone #1 represents a PEF cell clone with complete deletion of the coding region of the TMEM239 gene. (B) Genotype identification of SCNT-producing piglets. DNA was extracted from piglet ear margins for PCR identification. Primer pair F3/R4 was used to identify piglet genotypes. Primer pair F3/R4 amplified a 7678-bp band in wild-type piglets and a ~513-bp band in TMEM239 homozygous knockout piglets. The red arrow indicates the wild-type bands from piglet #355–4. WT, Wild-type piglet, NC, Negative control. (C) Gating strategy of the proliferation assay. 7-aminoactinomycin D (7-AAD) was utilized to identify live cells, while anti-pig CD45 was employed to isolate lymphocytes. Subsequently, based on CD45$^+$, cells were analyzed for the presence of CD3$^+$, Monocyte/Granulocyte (Mon/Gra), B cells, CD4$^+$CD8$^-$, CD4$^-$CD8$^+$, and CD4$^+$CD8$^+$ populations. (D

and E) Results of flow cytometry analysis on PBMCs from TMEM239$^{-/-}$ pigs and wild-type pigs. PBMCs from TMEM239$^{-/-}$ pigs and wild-type pigs were surface-labeled with anti-porcine CD45 (Pacific Blue), anti-porcine CD3ε-FITC, anti-porcine 172a-PE, anti-porcine CD21-PE, anti-porcine CD4-APC, anti-porcine CD8-FITC, and anti-porcine CD21-PE.
(TIF)

**S6 Fig. Fluorescence micrographs and flow cytometric analysis of PBMCs infected with ASFV.** (A) Fluorescence micrographs illustrating ad-7GD replication. WT PBMCs and TMEM239$^{-/-}$ PBMCs were individually infected with the ad-7GD strain at an MOI of 1. Fluorescence micrographs were collected at 72 hpi. WT-1, WT-2, and WT-3 were the numbers of the wild-type piglets selected for the current experiment. Piglets numbers 355-(1–4) and 393-(1–5) correspond to piglets born to surrogate sows #355 and #393, respectively. (B-D) Flow cytometric analysis of PBMCs infected with ASFV. WT PBMCs and TMEM239$^{-/-}$ PBMCs were individually infected with the ad-7GD strain at an MOI of 1 for 16 h. These PBMCs were surface-labeled with anti-porcine CD45 (Pacific Blue) and anti-porcine 172a-PE. Meanwhile, a gate was set to detect ASFV infected cells.
(TIF)

**S1 Table. Information of ASFV strains used in this study.**
(DOCX)

**S2 Table. The sequences of sgRNAs, siRNAs and primers used in this study.**
(DOCX)

**S3 Table. The swine genome-wide CRISPR knockout screen sgRNA library in current study.**
(XLSX)

**S4 Table. The swine genome-wide CRISPR knockout screen sgRNA cell library.**
(XLSX)

**S5 Table. The list of genes enriched after the screening process.**
(XLSX)

**S6 Table. Identification of TMEM239 binding proteins in the uninfected-virus group by use of mass spectrometry.**
(DOCX)

**S7 Table. Identification of TMEM239 binding proteins in the infected-virus group by use of mass spectrometry.**
(DOCX)

## Author Contributions

**Conceptualization:** Zhonghua Liu, Zhigao Bu, Dongming Zhao.

**Data curation:** Dongdong Shen, Guigen Zhang, Xiaogang Weng.

**Formal analysis:** Dongdong Shen, Yuanmao Zhu, Encheng Sun, Jiwen Zhang, Fang Li.

**Funding acquisition:** Zhigao Bu, Dongming Zhao.

**Investigation:** Dongdong Shen, Guigen Zhang, Xiaogang Weng, Renqiang Liu.

**Methodology:** Dongdong Shen, Guigen Zhang, Xiaogang Weng, Zhiheng Liu, Xiangpeng Sheng, Yuting Zhang, Yan Liu, Yanshuang Mu, Fang Li, Junwei Ge.

**Project administration:** Dongdong Shen, Zhigao Bu.

**Resources:** Changyou Xia, Zhonghua Liu, Zhigao Bu, Dongming Zhao.

**Supervision:** Zhigao Bu, Dongming Zhao.

**Validation:** Zhonghua Liu, Zhigao Bu, Dongming Zhao.

**Writing – original draft:** Dongdong Shen, Dongming Zhao.

**Writing – review & editing:** Dongdong Shen, Zhigao Bu, Dongming Zhao.

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
