## [Decision Letter · Decision Letter 0]

7 Jan 2024

Dear Prof. Zhao,

Thank you very much for submitting your manuscript "A genome-wide CRISPR/Cas9 knockout screen identifies TMEM239 as a crucial host factor in facilitating African Swine Fever Virus entry into early endosomes" for consideration at PLOS Pathogens. As with all papers reviewed by the journal, your manuscript was reviewed by members of the editorial board and by several independent reviewers. In light of the reviews (below this email), we would like to invite the resubmission of a significantly-revised version that takes into account the reviewers' comments.

We agree with the reviewers agreed that your manuscript presents interesting novel data relevant to understanding African swine fever virus interactions with host macrophages. However, several of the conclusions are not fully supported by the data presented.

Specific points raised by each reviewer are included in their review reports and must be addressed point by point. Some conclusions drawn by the authors are too strongly stated, for example the description of TMEM293 as crucial for virus infection implies an essential role whereas the data presented shows deletion or knock down reduces virus replication. The conclusion that the TREM293 host factor is required for virus entry into early endosomes is not supported by the data and additional experiments should be included to support this conclusion or alternative explanations discussed. Some points raised require including additional experimental controls. Reviewer 3 points out that in the siRNA experiments to knock down TMEM239 multiple siRNAs should be included as controls for off target effects. Likewise additional controls would need to be included to support the conclusion that reduced ASFV replication in PAMS from transgenic KO TREM239 piglets host factor is due to reduced virus entry into early endosomes rather than an indirect effect at other stages of replication. Reviewers 1 and 3 make suggestions for controls which could be included in support of this conclusion. Although proteomic data is included showing association of TREM239 (Fig 5) with virus protein p72 in uninfected cells this evidence is insufficient to conclude a role for the interaction in virus entry. To draw this conclusion evidence in the context of virus infection should be included.

Some methods and reagents used are not described in detail. Reviewers 1 and 2 point out that details, including characterization and justification of the use of several virus strains, are not included. Reviewer 1 points out the need to ensure purity of virus used in studies of entry and that methods used for virus preparation are not included in the manuscript. Reviewer 2 points out that some experiments were carried out at high moi with a gene-deleted virus and not with wild type virus or at a lower moi. Information on the phenotype of the TREM239 knock out pigs including endosomal function and details of antibodies used are also lacking.

We cannot make any decision about publication until we have seen the revised manuscript and your response to the reviewers' comments. Your revised manuscript is also likely to be sent to reviewers for further evaluation.

Sincerely,

Linda Kathleen Dixon

Academic Editor

PLOS Pathogens

Alison McBride

Section Editor

PLOS Pathogens

Kasturi Haldar

Editor-in-Chief

PLOS Pathogens

orcid.org/0000-0001-5065-158X

Michael Malim

Editor-in-Chief

PLOS Pathogens

orcid.org/0000-0002-7699-2064

We agree with the reviewers agreed that your manuscript presents interesting novel data relevant to understanding African swine fever virus interactions with host macrophages. However, several of the conclusions are not fully supported by the data presented.

Specific points raised by each reviewer are included in their review reports and must be addressed point by point. Some conclusions drawn by the authors are too strongly stated, for example the description of TMEM293 as crucial for virus infection implies an essential role whereas the data presented shows deletion or knock down reduces virus replication. The conclusion that the TREM293 host factor is required for virus entry into early endosomes is not supported by the data and additional experiments should be included to support this conclusion or alternative explanations discussed. Some points raised require including additional experimental controls. Reviewer 3 points out that in the siRNA experiments to knock down TMEM239 multiple siRNAs should be included as controls for off target effects. Likewise additional controls would need to be included to support the conclusion that reduced ASFV replication in PAMS from transgenic KO TREM239 piglets host factor is due to reduced virus entry into early endosomes rather than an indirect effect at other stages of replication. Reviewers 1 and 3 make suggestions for controls which could be included in support of this conclusion. Although proteomic data is included showing association of TREM239 (Fig 5) with virus protein p72 in uninfected cells this evidence is insufficient to conclude a role for the interaction in virus entry. To draw this conclusion evidence in the context of virus infection should be included.

Some methods and reagents used are not described in detail. Reviewers 1 and 2 point out that details, including characterization and justification of the use of several virus strains, are not included. Reviewer 1 points out the need to ensure purity of virus used in studies of entry and that methods used for virus preparation are not included in the manuscript. Reviewer 2 points out that some experiments were carried out at high moi with a gene-deleted virus and not with wild type virus or at a lower moi. Information on the phenotype of the TREM239 knock out pigs including endosomal function and details of antibodies used are also lacking.

Reviewer's Responses to Questions

**Part I - Summary**

Reviewer #1: Shen et al. have presented a well written manuscript exploring the role of host factors in supporting African swine fever replication. The authors use a CRISPR screen in WSL cells, a continuous cell line derived from wild boar lungs, to identify genes that are essential for ASFV replication and then perform a number of experiments to demonstrate the relevance of TMEM239 to virus replication in primary macrophages and understand the mechanism. Interestingly, the authors identify one gene that was identified by another group in a recent publication using the same cell line. The present study has similar problems to this other study in that the experiments to demonstrate relevance in primary porcine macrophages are less convincing. The authors strengthen their case by using a gene deleted pig, but the experiments with cells from these animals are not sufficiently controlled to support the authors conclusions. The experiments to prove a mechanism also require further data to fully support the authors conclusions. Finally the experiments rely on a series of adapted viruses which are not sufficiently characterized and the justification for using a particular virus in a particular experiment is absent. Although I think the authors have some very interesting data, I cannot recommend it for publication in its present state.

Reviewer #2: General Comments

African swine fever (ASF) is a fatal disease of pigs caused by the African swine fever virus (ASFV). Our limited understanding of its replication strategy and interaction with the host have hampered the development of ASFV mitigation strategies such as vaccines and antiviral compounds. In the manuscript entitled “A genome wide CRISPR/Cas9 knockout screen identifies TMEM239 as a crucial host factor in facilitating African Swine Fever Virus entry into early endosomes”, Shen and coworkers tried to identify host factors required for ASFV replication. Genome wide CRISPR knock-out (GeCKO) wild boar WSL lung cells targeting 20,580 pig genes were established and infected three times with a genotype II ASFV to select for resistant cells. The DNA of the surviving cells was analyzed for sgRNA enrichment as compared to uninfected GeCKO WSL cells. It was determined that TMEM239, SLD-DMB and Rab14 were enriched. Next, the role of the TMEM239 gene/protein and its associated Rab14 gene/protein in the ASFV replication cycle was determined by creating TMEM239 and Rab14 knock-out WSL cell; both knock-out WSL cells showed significantly less ASFV replication as compared to the wild-type WSL cells. Then, it was shown that the TMEM239 protein interacted with the early endosomal marker Rab5A and affected the colocalization of the viral capsid p72 protein with Rab5A. These results led to the conclusion that TMEM239 plays a role in the early stage of ASFV replication. Using siRNA knock-down studies of TMEM239 in PAM cells revealed a suppression of virus replication. In addition, ASFV replication was significantly reduced in TMEM239-/- peripheral blood mononuclear cells (PBMCs) from TMEM239-/- knock-out piglets. The authors conclude that their study identified a novel host factor, TMEM239, which is required for ASFV replication and facilitates ASFV entry into early endosomes; this work can also provide insights for the development of ASF-resistant pigs.

In the present manuscript, Shen and coworkers used CRISPR/Cas9 knock-out screens in wild boar WSL cells to identify a host factor (TMEM239) which seems critical for ASFV replication. This is a well written and interesting manuscript but several critical questions, especially on methodology, materials and execution still remain. Importantly, the use of a multitude of different ASFV isolates - some wild-type ASFVs, some gene-deleted and attenuated ASFVs – which are not well defined in the Materials and Methods section, in various experiments throughout the manuscript is rather confusing; it is not clear why this approach was chosen by the authors. Also, critical experiments were performed with a high MOI with a gene-deleted attenuated ASF virus (ad-7GD), and not with a wild-type ASF virus or with a much lower MOI. Also, it is not clear why the seven surviving TMEM239 knock-out piglets were not used for ASFV-infection experiments. This would be a critical proof for the role of TMEM239 in ASFV replication.

Reviewer #3: The authors describe a genome wide CRISPR survival screen which identifies host factors important for ASFV virus infection and replication. A similar approach was recently published by Pannhost et al and both studies identified the same top hit - SLA-DM. Here the authors go on to characterise two additional hits identified in their screen, TMEM239 and RAB14. After initial studies they focus on TMEM239 as this gave the clearest phenotype. Virus growth curve analysis showed significant reductions of ASFV replication in TMEM239 KO cells and following adsorption assays , proteomics, pulldowns and colocalisation, the authors suggest that KO of TMEM239 impedes entry into early endosomes. Finally TMEM239 transgenic KO piglets were generated. From the description KO of TMEM239 had significant impacts on the heath of the piglets. Despite this they isolated PAMs from the pigs and showed ASFV replication was reduced in the cells from the transgenic pigs when compared to wild type pigs. While some of the findings are compelling, many of the conclusions are not fully supported by the data presented.

**Part II – Major Issues: Key Experiments Required for Acceptance**

Reviewer #1: Use of three WSL adapted viruses that don’t appear to be characterized. What changes if any occurred during the adaptation to WSL cells and do these viruses replicate in primary macrophages? The 7-GD virus used for most of the experiments in WSL cells is missing the only confirmed viral component of the extracellular envelope (CD2v/pEP402R) and MGF genes known to be involved in host range and cell tropism.

The Hernaez paper (reference 37) clearly show that the routes of virus entry in Vero cells and macrophages differ depending on the purity of the virus preparation. The method of virus preparation is missing from the methods and should be included. In order to interpret their mechanistic data the authors need to define the methods of entry into WSL cells, this could be carried out following some of the methodology described in the Hernaez paper. This section also needs some data on the effect of TMEM239 deletion on normal endosomal function, are the authors showing a specific effect on ASFV or not? If the virus isn’t in Rab7+ endosomes, where is it?

The TMEM239-/- data is a strength of the paper, but additional data is required to support the authors conclusions. Any ASFV replication in PBMCs is mostly likely within monocytes and related cells (around 10% of the total cells), therefore the authors need to show that the PBMCs from the TMEM239-/- pigs have a similar number of ASFV susceptible cells and that the same cell types are being infected in their experiments.

Reviewer #2: Major Issues:

1. What was the purpose of using multiple ASFV strains, some being wild-type, some being gene-deleted attenuated viruses? Please provide detailed information on the abbreviations used for the ASFVs. It seems like the ad-7GD virus was used for many experiments described in the manuscript including the screening of the GeCKO WSL library (see Figure 1) and the testing of knock-out cells for ASFV replication (see figure 2). This reviewer believes that Ad-7GD is an attenuated, gene-deleted virus. Were the above-described experiments repeated with a wild-type virus with the same MOI?

2. Were the ASFVs used in this study sequenced? How were these viruses amplified and their respective titer obtained?

3. The authors custom designed approx. 186 000 sgRNA targeting more than 20,000 swine genes. They have not provided any of the sgRNA sequences, names of target genes or number of sgRNAs per gene. This information needs to be provided in supplementary files. It is not

---

## [Editor Report · Decision Letter 1]

28 Apr 2024

Dear Prof. Zhao,

Thank you very much for submitting your manuscript "A genome-wide CRISPR/Cas9 knockout screen identifies TMEM239 as a crucial host factor in facilitating African Swine Fever Virus entry into early endosomes" for consideration at PLOS Pathogens. As with all papers reviewed by the journal, your manuscript was reviewed by members of the editorial board and by several independent reviewers. The reviewers appreciated the attention to an important topic. Based on the reviews, we are likely to accept this manuscript for publication, providing that you modify the manuscript according to the review recommendations.

The points raised by reviewers have been addressed and the manuscript has been modified as described in the response to reviewers. Some further minor revisions to more thoroughly address some specific points are required, as described below, before the manuscript can be accepted.

Specific minor points to be addressed:

1. TMEM239 knockdown or knockout reduces replication of different ASFV strains by a relatively small amount, one or two logs maximum. Therefore, it is not accurate to describe TMEM293 as a “crucial host factor in facilitating ASFV entry into early endosomes”. The title should be reworded to better represent the results. For example, use the word “important” rather than “crucial”. This applies to other sections of the manuscript which describes TMEM293 as a crucial host factor for example the heading on line 283, heading of figure S2.

2. A major point raised by the reviewers was the inadequate description and rationale for use of the different ASFV strains used in the study. Explanation has been provided in Methods Cells and Viruses section of the characteristics of these strains including genes deleted from cell adapted or attenuated strains, how they were derived and cell tropism. Including a table or diagram to show the parental strain, how they were derived and which genes are deleted from different strains compared to the highly virulent wild type strain would help understanding. Accession numbers of full genome sequences could be included in the table.

3. A further point raised by reviewers was that many experiments were carried out using cell culture adapted or gene-deleted viruses. Thus it is possible that genome differences between isolates may account for variation in replication levels and may not necessarily be associated TMEM293 knockout. Although this is mentioned in the revised Discussion (starting line 638), it is important to further discuss the potential limitations of the study in regard to the use of attenuated viruses.

4. A previous publication demonstrated the importance of the Niemann-Pick C type 1 (NPC1) endosomal protein in ASFV entry and should be cited and discussed ( PLoS Pathog. 2022 Jan 26;18(1):e1009784. doi: 10.1371/journal.ppat.1009784).

Sincerely,

Linda Kathleen Dixon

Academic Editor

PLOS Pathogens

Alison McBride

Section Editor

PLOS Pathogens

Michael Malim

Editor-in-Chief

PLOS Pathogens

orcid.org/0000-0002-7699-2064

The points raised by reviewers have been addressed and the manuscript has been modified as described in the response to reviewers. Some further minor revisions to more thoroughly address some specific points are required, as described below, before the manuscript can be accepted.

Specific minor points to be addressed:

1. TMEM239 knockdown or knockout reduces replication of different ASFV strains by a relatively small amount, one or two logs maximum. Therefore, it is not accurate to describe TMEM293 as a “crucial host factor in facilitating ASFV entry into early endosomes”. The title should be reworded to better represent the results. For example, use the word “important” rather than “crucial”. This applies to other sections of the manuscript which describes TMEM293 as a crucial host factor for example the heading on line 283, heading of figure S2.

2. A major point raised by the reviewers was the inadequate description and rationale for use of the different ASFV strains used in the study. Explanation has been provided in Methods Cells and Viruses section of the characteristics of these strains including genes deleted from cell adapted or attenuated strains, how they were derived and cell tropism. Including a table or diagram to show the parental strain, how they were derived and which genes are deleted from different strains compared to the highly virulent wild type strain would help understanding. Accession numbers of full genome sequences could be included in the table.

3. A further point raised by reviewers was that many experiments were carried out using cell culture adapted or gene-deleted viruses. Thus it is possible that genome differences between isolates may account for variation in replication levels and may not necessarily be associated TMEM293 knockout. Although this is mentioned in the revised Discussion (starting line 638), it is important to further discuss the potential limitations of the study in regard to the use of attenuated viruses.

4. A previous publication demonstrated the importance of the Niemann-Pick C type 1 (NPC1) endosomal protein in ASFV entry and should be cited and discussed ( PLoS Pathog. 2022 Jan 26;18(1):e1009784. doi: 10.1371/journal.ppat.1009784).

Reviewer Comments (if any, and for reference):

Figure Files:

Data Requirements:

Reproducibility:

References:

---

## [Editor Report · Decision Letter 2]

13 May 2024

Dear Prof. Zhao,

We are pleased to inform you that your manuscript 'A genome-wide CRISPR/Cas9 knockout screen identifies TMEM239 as an important host factor in facilitating African swine fever virus entry into early endosomes' has been provisionally accepted for publication in PLOS Pathogens.

Best regards,

Linda Kathleen Dixon

Academic Editor

PLOS Pathogens

Alison McBride

Section Editor

PLOS Pathogens

Michael Malim

Editor-in-Chief

PLOS Pathogens

orcid.org/0000-0002-7699-2064

All points raised by reviewers have been addressed.
---

## [Editor Report · Acceptance letter]

12 Jul 2024

Dear Prof. Zhao,

We are delighted to inform you that your manuscript, "A genome-wide CRISPR/Cas9 knockout screen identifies TMEM239 as an important host factor in facilitating African swine fever virus entry into early endosomes," has been formally accepted for publication in PLOS Pathogens.

Best regards,

Michael Malim

Editor-in-Chief

PLOS Pathogens

orcid.org/0000-0002-7699-2064